# Additive effects of *Trichoderma* isolates for enhancing growth, suppressing southern blight and modulating plant defense enzymes in tomato

Nusrat Jahan Mishu[1], Md. Robiul Hasan[1], Shah Mohammad Naimul Islam[2], Jannatun Nayeema[1], Md. Motaher Hossain🔘[1]*

1 Department of Plant Pathology, Gazipur Agricultural University, Gazipur, Bangladesh, 2 Institute of Biotechnology and Genetic Engineering (IBGE), Gazipur Agricultural University, Gazipur, Bangladesh

\* hosssainmm@gau.edu.bd

## Abstract

Southern blight, caused by *Sclerotium rolfsii*, poses a significant economic threat to tomato cultivation. This study involved the isolation, characterization, and evaluation of three selected *Trichoderma* isolates (Tri2, Tri3, and Tri6), applied individually and in combination, for their potential to promote plant growth and suppress southern blight under both *in vitro* and *in vivo* conditions. These isolates exhibited multiple plant growth-promoting traits, including cellulase, protease, amylase, lipase, catalase, and phosphate-solubilizing activities. Furthermore, they performed as efficient antagonists, inhibiting the mycelial growth by up to 88.8% and the oxalic acid production of *S. rolfsii* by up to 81.9%. The *Trichoderma* isolates significantly enhanced tomato seed germination and seedling vigor ($p < 0.05$). In seed tray and pot, experiments, consortium treatments (dual and triple application) demonstrated significantly greater plant height (≤168.8%), biomass (≤507.3%), leaf number (≤150%), leaf diameter (≤86.2%), chlorophyll content (≤322%), stem diameter (≤129.1%), gas exchange parameters, and root colonization than control and single treatments ($p < 0.05$). Additionally, these consortium treatments exhibited significantly higher efficacy in reducing damping-off (≤92%) and southern blight severity (≤80%) caused by *S. rolfsii*, compared to untreated plants ($p < 0.05$). Biochemical analyses revealed that *Trichoderma*-treated plants challenged with *S. rolfsii* showed reduced oxidative stress, evidenced by lower hydrogen peroxide ($H_2O_2$) and malondialdehyde (MDA) levels. The treatments also increased osmoprotectant levels such as soluble sugars, proline, phenolics, and flavonoids, along with the activities of defense-related enzymes, including peroxidase (PO), polyphenol oxidase (PPO), and phenylalanine ammonia-lyase (PAL), compared to *S. rolfsii*-infected controls ($p < 0.05$). Under field conditions, treating *S. rolfsii*-inoculated plants with *Trichoderma* isolates, whether singly, in pairs, or as a trio, significantly increased plant height, yield, and fruit Brix content ($p < 0.05$). The consortium application (Tri2 + Tri3 and Tri2 + Tri3 + Tri6) led to

**Data availability statement:** All relevant data are within the manuscript and its Supporting Information files.

**Funding:** The author(s) received no specific funding for this work.

**Competing interests:** The authors have declared that no competing interests exist.

the highest increases in plant height (≤94%), fruit number (≤114%), yield (≤19.59 t/ha), and Brix (≤4.88). These findings suggest that the additive interactions among *Trichoderma* isolates enhance tomato growth and suppress *S. rolfsii*, offering an eco-friendly and effective strategy for managing southern blight.

## Introduction

Tomato (*Solanum lycopersicum* L.) is a popular and commonly consumed vegetable worldwide. It belongs to the *Solanaceae* plant family, which includes several other well-known vegetable plants. Tomato fruits are known to have an elevated level of phenolics, which have both preventive and therapeutic properties against a range of diseases, including cancer [1]. Although tomato is typically grown in winter in Bangladesh, it is now cultivated year-round. However, tomato production faces many challenges, including an array of biotic and abiotic stresses. Plant diseases are a significant biotic constraint for tomato cultivation worldwide, negatively impacting yield and quality. Plant diseases, excluding viruses, can cause yield losses of up to 40% in tomato crops when no crop protection is used [2].

Tomato plants are attacked by a number of important fungal pathogens, resulting in substantial hindrances to growth and yield. One of the most severe fungal threats is southern blight disease, caused by *Sclerotium rolfsii* (syn. *Athelia rolfsii*), a soilborne pathogen responsible for significant economic losses in tomatoes and other field-grown crops worldwide. The disease was first reported in tomato plants in 1892 [3] and is now recognized as a destructive pathogen infecting over 500 plant species, particularly in warm climates [4]. Depending on the host plant species, southern blight is also known as *Sclerotium* root rot, collar rot, southern stem rot, or cottony white rot. The fungus can remain in the soil or crop residues as sclerotia for years and infect plants at various stages of growth. The soilborne inoculum ceases seed germination and causes damping off, leading to seedling mortality [5]. Necrotic rot may also result from the fungus infecting the stem at or close to the soil line [6]. The stem is encircled by quickly developing lesions, which cause sudden and irreversible wilting of the plant [7]. When fruits touch the soil, they might become infected and rot. The fungus can cause significant damage to crops under favourable soil and weather conditions, often resulting in devastating losses [8]. For instance, it led to an 80% reduction in *Arachis hypogaea* yields in the United States, resulting in economic losses totaling $36.8 million [9].

The control strategy of *S. rolfsii* is highly challenging due to the soilborne nature of the pathogen. The typical recommended preventive strategies for the pathogen involve eradicating infected plants as sources of inoculum, using fungicides to treat plants and seeds, and adding non-host plants in crop rotations such as black oat [10–12]. Of these various approaches, fungicides have been considered the principal management technique for southern blight [13]. However, the fungicidal control of soilborne diseases is generally expensive and met with limited success [14]. Excessive use of fungicides is also associated with ecotoxicological risks in non-target aquatic systems, the environment, and public health [15]. As a result, the need to identify and use control strategies other than traditional fungicides is demanding.

Recently, biological control of soilborne phytopathogens has been the center of growing attention for in-depth studies due to their high efficacy against soilborne pathogens and non-toxic environmental effects. Among biocontrol agents, some *Trichoderma* isolates are well-known for their ability to suppress the growth and spread of harmful phytopathogenic fungi [16–18]. They are commonly employed as biocontrol agents to improve plant growth and combat pathogens in greenhouses and fields [19–20]. A variety of *Trichoderma* isolates has already demonstrated efficacy against *S. rolfsii* in various crops [21–22]. Their biocontrol mechanisms include mycoparasitism, competition for nutrients and space, antibiosis, and induced plant resistance [16,23]. Notably, *Trichoderma* modulates enhanced plant defense responses through the activation of key enzymes such as peroxidase (PO), β-1,3-glucanase, phenylalanine ammonia-lyase (PAL), polyphenol oxidase (PPO), chitinase, and superoxide dismutase (SOD), along with proline accumulation [24]. The induction of antimicrobial enzymes is pivotal in conferring plant resistance against pathogens [25]. These mechanisms highlight the potential of *Trichoderma* as a targeted strategy for managing southern blight.

The southern blight fungus *S. rolfsii* is notorious for its ability to produce oxalic acid, a key virulence factor that plays a pivotal role in disease progression. Oxalic acid not only facilitates host tissue colonization but also creates acidic microenvironments, enabling the pathogen to attack a wide range of host plants with devastating effects [26]. Although some studies have investigated the use of *Trichoderma* in managing *S. rolfsii* in various crops in different countries [27–28], limited research globally has directly focused on inhibiting oxalic acid production as a targeted biocontrol mechanism. Targeting this virulence mechanism by inhibiting oxalic acid production through the application of *Trichoderma* has emerged as a promising and novel strategy for managing this destructive disease. Additionally, studies suggest that applying *Trichoderma* strains in combination is more effective than using single strains for controlling soilborne diseases [29–31]. In Bangladesh, several studies have documented the positive role of *Trichoderma* isolates in controlling *S. rolfsii* in crops like soybean and carrot [32–34]. However, these studies have neither explored potential biocontrol mechanisms of disease suppression by *Trichoderma* nor evaluated the additive effects of multiple *Trichoderma* strains in achieving maximum biocontrol efficacy. Comparable studies have demonstrated the effectiveness of *Trichoderma* consortia in suppressing *S. rolfsii* in groundnut in India. For instance, Ayyandurai et al. [28] showed that a combination of *T. longibrachiatum* and *T. asperellum*, in association with mahua oil cake, significantly reduced stem rot incidence in groundnut through the production of antimicrobial metabolites and volatile organic compounds. While this study established the potential of *Trichoderma* consortia in groundnut, comparable mechanistic insights and additive evaluations are lacking for tomato crops, particularly in the context of oxalic acid inhibition by *Trichoderma* isolates.

Hence, the present study aimed to address these gaps by isolating and characterizing oxalic acid-inhibiting *Trichoderma* strains and evaluating various combinations for their potential to promote plant growth, enhance yield, and suppress *S. rolfsii* in tomato crops. Additionally, the biochemical mechanisms underlying disease suppression in *S. lycopersicum* are investigated to provide a comprehensive understanding of *Trichoderma*-mediated biocontrol efficacy.

## Materials and methods

### Collection of soil samples for *Trichoderma* isolation

Soil samples were collected from the root zones (at a depth of 15 cm) of healthy plants, including soybean, tomato, cucumber, eggplant, and peanut, in the Gazipur, Chandpur, Noakhali, and Lakshmipur districts of Bangladesh. For each plant species, three rhizospheric soil samples were collected per location to ensure replication and consistency in isolate recovery. These samples were then stored at 4°C in the laboratory for subsequent analysis.

### Pathogen, plant material, and potting medium

The plant pathogen *Sclerotium rolfsii* SR-1 used in this study was sourced from the stock cultures of the department [6]. The fungal cultures were stored on potato dextrose agar (PDA) slants at 4°C until needed. Tomato cv. Mintoo Super F1 Year-round (Lal Teer Seed Company, Gazipur, Bangladesh) was selected as the host plant for the research. The soil

collected from the university research field was used as the potting medium [35]. It was sandy loam with a pH of 6.38, containing 1.08% organic carbon, 1.87% organic matter, 0.27% nitrogen, 0.09% phosphorus, and 0.87% potassium. The soil was autoclaved twice, 24 hours apart, at 121°C and 15 psi for 20 minutes each.

### Isolation of *Trichoderma*

A serial dilution technique was employed to isolate the *Trichoderma*. The initial dilution was made by mixing 1 g of soil with 9 mL of sterile distilled water (SDW). Then, 100 µL of this dilution was spread onto PDA plates supplemented with streptomycin sulfate (100 ppm) (Nacalai Tesque, Inc., Kyoto, Japan). After incubating for four days at 25°C, individual fungal colonies identified as *Trichoderma* were isolated and purified for further study.

### Screening *Trichoderma* for antagonistic activity against *Sclerotium rolfsii*

**Dual culture assay.** To test the antagonistic activity of ten *Trichoderma* isolates, mycelial agar discs of *S. rolfsii* and each isolate were positioned on opposite sides of a PDA plate. A positive control plate was amended with the fungicide Provax-200 (Hossain Enterprise C.C. Limited, Dhaka, Bangladesh) at 200 ppm, while a negative control plate was left without fungal inoculation or fungicidal amendment. The plates were cultured at 28°C for five days, and the fungal growth was recorded when the control *S. rolfsii* plates covered the entire plate. The percent inhibition (PI) of *S. rolfsii* mycelia by *Trichoderma* was calculated [36]. The approaching hyphae were also examined and photographed under a light microscope for their morphological characteristics.

**Culture filtrate assay.** The effectiveness of *Trichoderma* culture filtrates against the *S. rolfsii* isolate was evaluated [37]. *Trichoderma* was cultured in potato dextrose broth (PDB), and the resulting broth was filtered using a 0.45 µm filter. Since the undiluted *Trichoderma* culture filtrates completely inhibited *S. rolfsii* mycelial growth, cell-free filtrates were diluted to assess differences in effectiveness among various *Trichoderma* strains. The filtrates were diluted to 10%, 20%, and 30% (v/v) to create a gradient that could distinguish the relative antifungal potency of the isolates while avoiding total suppression at higher doses. The filtrates were diluted accordingly and applied to PDA plates, with PDA treated with Provax-200 (at 200 ppm) serving as a positive control. A mycelial plug of *S. rolfsii* was then placed on each plate. After incubation at 28°C for five days, the inhibition percentage of mycelial growth was subsequently calculated [36].

### Selection, identification, and characterization of elite antagonists

For molecular characterization, three *Trichoderma* isolates Tri2, Tri3, and Tri6 were selected based on their superior antagonistic activity (over 70% inhibition of *S. rolfsii* mycelial growth) observed in preliminary *in vitro* dual culture and culture filtrate assays. Plant DNA Kit (Promega) was used to extract genomic fungal DNA. The ITS region was amplified by polymerase chain reaction (PCR) using ITS4/ITS5 primers [38]. After purification (Promega, Madison, WI, USA), PCR products were sent to Macrogen, Korea, for sequencing. Geneious Prime v2023.1.2 was used to edit the sequences, and the Basic Local Alignment Search Tool (BLAST) program was utilized to identify closely related sequences.

### *Trichoderma* beneficial trait characterization

The modified Pikovskaya's (PVK) agar medium was used to assess phosphate solubilization activity [39]. A translucent halo zone around the colony was indicative of phosphate solubilization. To conduct the cellulase assay, Czapek-Mineral Salt Agar with the addition of Carboxy Methyl Cellulose (CMC) at 5.00 g/L was used for culturing *Trichoderma* [39]. The development of an opaque halo area around the colonies was used to observe cellulase production. *Trichoderma* was cultured on Starch Agar Medium to assess amylase activity [40]. The presence of a clear, hyaline zone around the colony specified positive activity. For protease screening, a Casein Agar Medium was used to cultivate the *Trichoderma* isolates, as per the method of Vijayaraghavan and Vincent [41]. The appearance of a colorless zone identified proteolytic activity.

The lipase production by *Trichoderma* was assessed by using the procedure of Hankin and Anagnostakis [40]. The catalase test was carried out using the method with Hossain and Sultana [39], where the presence of catalase was indicated by the copious bubbles liberated in the hydrogen peroxide.

### Estimation of oxalic acid inhibition by *Trichoderma*

The inhibition of oxalic acid production by *Trichoderma* was estimated by co-culturing *S. rolfsii* and *Trichoderma* in PDB (ten 5 mm mycelial discs of each in 100 mL). PDB solely inoculated with the pathogenic fungus *S. rolfsii* acted as the control. Each culture had three replicates. After incubation at 25 ºC for two weeks, cultures were centrifuged at 4000 rpm for 25 minutes. Five milliliters of each collected supernatant were placed in a sterilized 15 mL centrifuge tube, followed by the addition of 4 mL $CaCl_2$-Acetate buffer [42]. The mixtures underwent centrifugation at 4000 rpm for 15 minutes, after which supernatants were discarded and deposits were rinsed with 5 mL acetic acid (5%) saturated with calcium oxalate and re-centrifuged. Each deposit was solubilized in 4 N $H_2SO_4$ and heated in a water bath at 80–90°C. Heated solutions were filtered with 0.02 N $KMnO_4$ until a light pink hue appeared. Oxalic acid quantification was done by considering that 1 mL of 0.02 N $KMnO_4$ reacted with 1.2653 mg of oxalic acid [42].

### Preparation of *Trichoderma* and *S. rolfsii* inocula

To prepare the fungal inoculum, 100 g of wheat grains were soaked in 100 mL of water in a 500 mL Erlenmeyer flask, autoclaved, and subsequently inoculated with 30 mycelial disks (5 mm in diameter) excised from the actively growing margin of 7-day-old fungal cultures grown on PDA [43]. The flask was incubated at 25°C for 15 days, with gentle shaking every three days to ensure uniform and complete colonization of the wheat grains by *Trichoderma*. After incubation, the colonized grains were spread on paper towels in a tray and air-dried for several days. The dried wheat grain inoculum was then stored in a plastic container at 4°C until further use.

### *In vitro* effect of selected fungus isolates on the germination and growth of tomato plants

*Trichoderma* isolates (Tri2, Tri3, and Tri6) were cultured on PDA for one week at 25°C, and the spores were collected in sterilized saline (0.85% NaCl). The harvested spore suspension was filtered through sterilized muslin cloth to create a stock suspension; one milliliter of the spore suspension was combined with nine milliliters of saline. The optical density ($OD_{600}$) 1.00 (SHIMADZU Corporation, Japan) corresponded to a spore concentration of $1 \times 10^5$ spores/mL, determined using a hemocytometer. To surface sterilize tomato seeds, seeds were first washed with 20 mL SDW, and then immersed in a 1.5% NaOCl solution for 5 minutes. After sterilization, seeds were rinsed twice with 20 mL SDW and dried by blotting. Surface-sterilized seeds were dipped in *Trichoderma* spore suspension for 1–2 minutes. The untreated control seeds were dipped in water. For the germination test, 50 tomato seeds were taken in each replication and placed on two sheets of wet sterile filter paper (Whatman No. 1) in a 9.0 cm Petri plate and incubated at 25°C for seven days and data on germination percentage and seedling length were taken and vigor index was determined [6]:

$$Vigor\ index\ =\ \%\ germination\ \times\ total\ plant\ length$$

### *In vivo* test for the growth promotion of the tomato plant by *Trichoderma*

**In seed trays.** Different combinations of *Trichoderma* isolates (Tri2, Tri3, and Tri6) were tested to study their ability to boost tomato plant growth in seed trays containing sterilized field soil. The experiment comprised eight treatment combinations, each with three replications, as shown in Table 1. Wheat grain inoculum of *Trichoderma* was (Tri2, Tri3, and Tri6) added to the potting soil at concentrations of 2.0%, 1.0%, and 0.7% (w/w) for single, dual, and triple combinations, respectively. These rates were based on prior studies [43] and optimized in preliminary trials to ensure effective

**Table 1. Treatments used in seed tray and pot experiments to evaluate the efficacy of single, dual, and triple combinations of *Trichoderma* for enhancing the growth of tomato plants.**

| Treatment | Composition |
|---|---|
| T1 (Control) | Uninoculated plants |
| T2 (Tri2) | Plants inoculated with *Trichoderma* isolate Tri2 |
| T3 (Tri3) | Plants inoculated with *Trichoderma* isolate Tri3 |
| T4 (Tri6) | Plants inoculated with *Trichoderma* isolate Tri6 |
| T5 (Tri2 + Tri3) | Plants inoculated with *Trichoderma* isolates Tri2 + Tri3 |
| T6 (Tri2 + Tri6) | Plants inoculated with *Trichoderma* isolates Tri2 + Tri6 |
| T7 (Tri3 + Tri6) | Plants inoculated with *Trichoderma* isolates Tri3 + Tri6 |
| T8 (Tri2 + Tri3 + Tri6) | Plants inoculated with *Trichoderma* isolates Tri2 + Tri3 + Tri6 |

colonization and plant response while avoiding excessive microbial load. Sterilized soil supplemented with autoclaved wheat grain was used as the control. The seed trays (15 cm × 15 cm) were filled with treated and untreated soil (300 g soil/ treatment), and 54 seeds were sown for each treatment. Ten days after sowing, data on the germination percentage were taken. After thinning, an equal number of seedlings was maintained for all treatments. Plants were grown for four weeks, watered every alternate day, and kept in a growth room at 24°C and 16/8 h photoperiod with an intensity of 650 µmol/ $m^2$/s (Sultana and Hossain, 2022) [6]. Leaf chlorophyll content was estimated by collecting fresh leaves (0.1 g) from three plants of each replication, homogenizing them with an 80% (v/v) acetone solution (1.5 mL), and centrifuging at 11,500 rpm for 10 minutes at 4°C. Absorbances of the supernatant were read at 663, 645, and 470 nm (UV-1800, SHIMADZU Corporation, Japan) to quantify chlorophyll *a* (*Chla*), chlorophyll *b* (*Chlb*), total chlorophyll, and carotenoids [44]. The entire plant was uprooted from the soil and cleaned using running tap water to eliminate any soil attached to the roots. Data were recorded on length, leaf number, and fresh weight of the whole seedling.

**In potted tomato plants.** The impact of single, dual, and triple combinations of *Trichoderma* isolates (Tri2, Tri3, and Tri6) on various growth parameters of tomato plants was evaluated in a pot study. The experiment comprised the same treatments used in seed tray experiments (Table 1). *Trichoderma* wheat grain inoculum (Tri2, Tri3, and Tri6) was added to the sterilized potting soil at concentrations of 2.0%, 1.0%, and 0.7% (w/w) of soil weight for single, dual, and triple combinations, respectively. Soil mixed with autoclaved wheat grain was used as the control. Each treatment was prepared in ten pots (11.50 cm × 15.0 cm). Each pot was filled with potting soil (500 g soil/pot). Two-week-old seedlings, raised in autoclaved soil, were transplanted into each pot. Plants were grown for seven weeks in the growth room under similar light conditions as described in the previous section and watered every alternate day. The *in-situ* chlorophyll (SPAD value) content was measured weekly, starting from two weeks after transplanting and continuing until six weeks after transplanting, with each SPAD value being the average of 10 readings [45]. Gas exchange factors, including stomatal conductance to $H_2O$ (gs), photosynthetic rate (Pn), leaf temperature (LT), and transpiration rate (E), were measured with an LI-6400XT system (LI-COR Biosciences, Nebraska, USA) under full sunlight from 10:00 AM to 11:30 PM seven weeks after transplanting. The plants were harvested to assess plant growth parameters, including shoot length, leaf diameter, full-grown leaflet number, shoot weight, root length, and root weight.

## Root colonization by *Trichoderma*

Research was conducted to assess the root colonization capacity of three *Trichoderma* isolates in individual and combined applications in tomato plants in pots. Plants were grown as mentioned earlier; the roots were collected at 2-, 4-, 6-, and 7-weeks post-transplantation. These time points were selected to monitor dynamic changes in *Trichoderma* root colonization over the full experimental period and align with stages of early seedling development, vegetative growth, and disease onset. Collected roots were thoroughly washed with water, sliced, weighed, and sterilized on the surface with a

5% (v/v) NaOCl for 1 minute, followed by three washes in SDW [46]. Root tissues from each sample were homogenized in SDW in a sterilized mortar and pestle. After filtering through two layers of sterilized cheesecloth, the homogenized suspension was diluted ten-fold. A 100 µL aliquot from each dilution was pipetted and spread on three PDA plates. After incubating at 25 ºC for 3–4 days, colonies were enumerated, and the population was expressed as colony-forming units (CFU)/gram fresh root weight.

## Suppression of disease by *Trichoderma* in seed tray

Suppression of *S. rolfsii* damping-off in tomato seedlings by *Trichoderma* isolates Tri2, Tri3, and Tri6 was assessed in seed trays. The experiment involved nine treatments with three replications for each (Table 2). T1 included only the inoculum of *S. rolfsii* (SR), while T9 included treatment with the fungicide Provax-200 (fungicide-protected) as a positive control. Other treatments included various combinations of treatment with *Trichoderma* isolates. Soil was mixed with *S. rolfsii*-colonized wheat grains at 2.0% (w/w), and *Trichoderma* wheat grain inoculum at the same optimized rates described earlier, based on the respective treatments [43]. Seed treated with Provax-200 (0.3% w/w) was considered a fungicide-protected check. Seeds were sown in seedling trays (15.0 cm × 15.0 cm) containing 300 g of soil mix in triplicate according to treatments. The trays were kept in a culture room (24°C, 16/8 h photoperiod). Damping-off incidence was recorded weekly over 4 weeks, based on the typical disease progression timeline for *S. rolfsii* in tomato [6]. The following formula was used to calculate disease incidence [43]:

$$\% \text{ Seedlings with damping off} = \frac{(No.\ of\ seedlings\ with\ damping\ off) \times 100}{(Total\ no.\ of\ seedlings\ in\ the\ tray)}$$

Additionally, data on sclerotia occurrence per $cm^2$ soil surface area were taken.

## Suppression of southern blight disease by *Trichoderma* in potted plant assay

The capability of *Trichoderma* isolates Tri2, Tri3, and Tri6 in single, dual, and triple combinations to suppress southern blight disease in potted tomato plants was tested. The experiment comprised nine treatments, with one being solely *S. rolfsii* (SR) and others with *Trichoderma* inoculum added at different concentrations (Section 2.11 and Table 2). The soil was prepared and treated with wheat grain inoculum of *S. rolfsii* and *Trichoderma* as described before. Each treatment was prepared in ten pots with 2-week-old seedlings uprooted from seed trays and transplanted into each pot. Provax-200 was used as a fungicide-protected check for T9 pots. All pots were placed in the culture room (24°C, 16/8 h

**Table 2. Treatments used in seed tray, pot, and field experiments to evaluate the efficacy of single, dual, and triple combinations of *Trichoderma* for controlling southern blight disease (*Sclerotium rolfsii*) in tomato.**

| Treatment | Treatment composition |
|---|---|
| **T1 (SR)** | Plants inoculated with *Sclerotium rolfsii* |
| **T2 (Tri2 + SR)** | Plants inoculated with *Trichoderma* isolate Tri2 + *S. rolfsii* |
| **T3 (Tri3 + SR)** | Plants inoculated with *Trichoderma* isolate Tri3 + *S. rolfsii* |
| **T4 (Tri6 + SR)** | Plants inoculated with *Trichoderma* isolate Tri6 + *S. rolfsii* |
| **T5 (Tri2 + Tri3 + SR)** | Plants inoculated with *Trichoderma* isolate (Tri2 + Tri3) + *S. rolfsii* |
| **T6 (Tri2 + Tri6 + SR)** | Plants inoculated with *Trichoderma* isolate (Tri2 + Tri6) + *S. rolfsii* |
| **T7 (Tri3 + Tri6 + SR)** | Plants inoculated with *Trichoderma* isolate (Tri3 + Tri6) + *S. rolfsii* |
| **T8 (Tri2 + Tri3 + Tri6 + SR)** | Plants inoculated with *Trichoderma* isolate (Tri2 + *Tri*3 + Tri6) + *S. rolfsii* |
| **T9 (Provax-200 + SR)** | Plants treated with Fungicide Provax-200 + inoculated with *S. rolfsii* |

photoperiod) for seven weeks, with disease development on each plant rated every week. A 0–5 scale was used to rate disease development [47]. When plants were four weeks old, leaves were collected, soaked in liquid nitrogen, and analyzed for biochemical parameters. The timing of disease assessments and biochemical analyses was selected based on the typical time frame for *S. rolfsii* symptom development and effective root colonization by *Trichoderma* in tomato plants [6,43].

**Quantification of hydrogen peroxide ($H_2O_2$) and malondialdehyde (MDA).** Collected leaves (0.1 g) were homogenized with 0.1% trichloroacetic acid (TCA). The resulting mixtures were centrifuged (11,500 rpm, 15 minutes, 4°C). The collected supernatant was utilized to quantify the levels of $H_2O_2$ and MDA [48–49].

**Quantification of proline and soluble sugar content.** The quantification of proline content was done following the method of Bates et al. [50]. Similarly, the valuation of soluble sugars (TSS) in tomato leaf tissues was performed following the techniques recommended by Dubois et al. [51].

**Quantification of total phenolic and flavonoid content.** The quantification of total phenolic and flavonoid contents in plant samples was carried out using colorimetric methods. For total phenolic content, the Folin-Ciocalteu method, as described by Ainsworth and Gillespie [52], was employed. In this approach, 0.1 g of lyophilized tomato leaf samples was first ground and mixed with 1.5 mL of 99.8% methanol. The mixture was then centrifuged at 15,000 rpm for 15 minutes at 4°C to obtain a clear extract. A 0.5 mL aliquot of this methanol extract was combined with 2 mL of Folin-Ciocalteu reagent, diluted 1:10 with deionized water, and 4 mL of a saturated Na2CO3 solution. After incubating the mixture at room temperature for 30 minutes with occasional stirring, the absorbance was measured at 765 nm using a spectrophotometer (Shimadzu Corporation, Japan). Methanol served as the blank. The total phenolic content was then calculated using a standard curve prepared using gallic acid.

For the quantification of total flavonoid content, the method outlined by Zhishen et al. [53] was used. Here, 1 mL of the methanol extract was mixed with 1.4 mL of water and 300 µl of 5% $NaNO_2$. The mixture was allowed to incubate for 5 minutes, after which 300 µl of 10% $AlCl_3$, 2 mL of 1 M NaOH, and 5 mL of water were added. The absorbance of the solution was recorded at 415 nm. The flavonoid content of the sample was then determined by comparing the measured absorbance to the calibration curve using standard catechin solutions.

## Quantification of polyphenol oxidase (PPO) and Peroxidase (PO) activities

PPO and PO activities were measured following the methods outlined by Whetten and Sederoff [54] and Hemeda and Klein [55]. To determine PPO activity, 0.5 g of leaves was first homogenized in 2 mL of 0.1 M Na-acetate buffer (pH 5.2), and then centrifuged at 15,000 rpm for 20 minutes at 4°C. The resulting supernatant was used for both PPO and PO activity measurements.

For PPO activity, 100 µl of the enzyme extract was added to a reaction mixture consisting of 0.1 M phosphate buffer (pH 7.0) and 120 mM catechol. The total volume of the solution was adjusted to 3.0 mL using distilled water. A blank sample was prepared by replacing the enzyme extract with water. The PPO activity was then measured by monitoring the absorbance at 420 nm using a spectrophotometer set to 30°C.

A similar procedure was followed to assess PO activity. A 100 µL aliquot of the enzyme extract was mixed with 1.68% guaiacol, 0.1 M phosphate buffer (pH 7.0), and 1% $H_2O_2$. The total volume was adjusted to 3 mL with distilled water, and a blank sample was prepared by replacing the enzyme extract with water. The PO activity was measured by tracking the change in absorbance at 470 nm using the spectrophotometer at 30°C. The enzyme activities for both PPO and PO were expressed as changes in absorbance per minute per milligram of protein (OD min$^{-1}$ mg protein$^{-1}$) [56].

## Quantification of phenylalanine ammonia-lyase (PAL) activities

PAL activity was quantified following the procedure outlined by Sadasivam and Manickam [57]. To begin, 0.5 g of leaves was homogenized in 2 mL of an extraction solution, which included 50 mM Tris-HCl buffer (pH 8.8), 15 mM β-mercaptoethanol, 5 mM EDTA, 5 mM ascorbic acid, 10 mM leupeptin, 1 mM phenylmethylsulfonyl fluoride (PMSF), and

0.15% w/v polyvinylpyrrolidone (PVP). The homogenate was filtered through cheesecloth and then centrifuged at 15000 rpm for 20 minutes at 4°C. The supernatant was collected, and 0.5 mL was mixed with the reaction mixture (16 mM L-phenylalanine, 50 mM Tris-HCl buffer, 3.6 mM NaCl. The mixture was incubated at 37°C for 1 hour, after which the reaction was stopped by adding 6 M HCl. The absorbance was measured at 290 nm. One unit of PAL activity was defined as the amount of enzyme required to produce 1 mmol of cinnamic acid per hour.

### Field assay

The experiment aimed to evaluate the effectiveness of *Trichoderma* isolates Tri2, Tri3, and Tri6, alone or in combination, in suppressing southern blight disease and improving yield in field conditions. The study design included nine treatments (Table 2). The treatments were laid out following a randomized complete block design (RCBD) with four replications. The plots were 2 m x 2 m with 0.5 m spacing in between. The soil was deep ploughed and fertilized with the recommended dose of fertilizers [6]. According to the experimental design, *S. rolfsii* and *Trichoderma* wheat grain inoculum were applied at the rate of 90 g m$^{-2}$, and mixed with soil. In dual and triple combinations of *Trichoderma*, an equal proportion of inoculum was taken from each isolate. Control plots did not receive *Trichoderma* inoculum; only the pathogen's inoculum was applied. Seedlings raised in the seed trays (20 × 12 × 5 cm) were uprooted and transplanted to the field. Nine plants per plot were assigned, with a spacing of 50 cm between rows and plants. Soils were drenched with Provax 200 (0.2% w/v) as positive checks [6]. Southern blight disease on each treatment was rated weekly using a 0–5 scale [47]. The *in-situ* chlorophyll (SPAD value) content was measured weekly, starting from two weeks after transplanting and continuing until six weeks after transplanting, with each SPAD value being the average of 10 readings [45]. Data was collected on the plant height and fruit brix content. The fruits were harvested in multiple intervals, and the yield was calculated accordingly. The data were recorded, and the mean values were calculated.

### Statistical analysis

The data obtained were analyzed using a one-way analysis of variance (ANOVA) to evaluate the results. Significant differences between treatments were decided by applying the least significant difference (LSD) test, with a significance level set at $p < 0.05$. The analysis was accomplished using Statistix 10 software.

## Results

### Isolation of *Trichoderma* from different locations

A total of 10 *Trichoderma* isolates were obtained from the rhizosphere soil of different crops in different locations in Bangladesh. Two isolates were found from the tomato, peanut, and eggplant rhizosphere, and three were from the soybean rhizosphere. One isolate was taken from the cucumber rhizosphere. The highest number of fungal isolates was found in Noakhali and Lakshmipur, giving 4 and 3 *Trichoderma* isolates, respectively. In contrast, two *Trichoderma* isolates were obtained from Gazipur, and a single isolate was obtained from Chandpur (S1 Table).

### Screening of *Trichoderma* isolates for antagonistic activity against *Sclerotium rolfsii*

**Dual culture assays.** The study assessed the antagonistic effects of 10 different *Trichoderma* isolates against the pathogenic fungus *S. rolfsii* in dual culture assays. In the control plates, *S. rolfsii* showed rapid growth, forming large colonies with an average diameter of 9.00 cm. However, when *S. rolfsii* was cultured alongside *Trichoderma*, its growth was significantly inhibited, with colony diameters ranging from 1.50 to 6.22 cm (Table 3). A clear inhibition zone was observed where the *Trichoderma* colonies interacted with *S. rolfsii*, and this zone remained consistent throughout the experiment (Fig 1). The inhibition of *S. rolfsii* mycelial growth ranged from 30.93% to 83.33%. The isolates Tri2, Tri3, and Tri6 were particularly effective, with Tri2 achieving the highest inhibition (83.33%), followed by Tri3 (82.22%) and

**Table 3. Inhibition of mycelial growth of *S. rolfsii* by different *Trichoderma* isolates in dual culture and culture filtrate assays.**

| Fungal inhibitor | Radial growth of the *S. rolfsii* in a dual culture plate | Inhibition of mycelial growth of *S. rolfsii* in dual culture (%)** | Inhibition of mycelial growth of *S. rolfsii* at different concentrations of culture filtrate of *Trichoderma* (%) | | |
|---|---|---|---|---|---|
| | | | 10 (%) | 20 (%) | 30 (%) |
| Tri1 | 4.40±0.40e* | 50.00±4.50e | 26.03±2.99f | 26.48±3.20fg | 29.34±0.06ef |
| Tri2 | 1.44±0.25h | 83.33±2.78b | 55.19±0.49b | 66.18±0.49b | 88.78±0.33b |
| Tri3 | 1.58±0.21g | 82.22±2.28bc | 44.67±0.33c | 51.74±0.34c | 85.22±0.13b |
| Tri4 | 5.40±0.49d | 39.07±5.47d | 31.65±1.80d | 33.33±3.39de | 34.32±3.51d |
| Tri5 | 6.10±0.08c | 31.67±0.85de | 26.5±1.38ef | 26.65±1.32d | 30.74±1.25fg |
| Tri6 | 2.46±0.03f | 72.22±0.32c | 25.07±0.04f | 36.11±0.00d | 72.52±0.15c |
| Tri7 | 6.18±0.26bc | 31.48±2.88de | 30.19±0.10e | 30.33±0.13ef | 30.48±0.13e |
| Tri8 | 6.17±0.12bc | 31.48±1.34de | 25.26±0.10f | 25.32±0.08g | 32.75±1.20g |
| Tri9 | 4.49±1.68e | 45.55±5.01e | 15.90±1.68c | 18.66±1.20h | 30.65±1.20 hg |
| Tri10 | 6.21±0.17b | 30.93±1.94de | 18.00±1.45f | 23.67±0.33g | 25.33±0.33g |
| Provax 200 | 0.00±0.00i | 100.00±0.00a | 100.00±0.00a | 100.00±0.00a | 100.00±0.00a |
| Control | 9.00±0.00a | 0.00±0.00f | 0.00±0.00g | 0.00±0.00i | 0.00±0.00h |

Note: *Trichoderma* isolates (Tri1 to Tri10) were evaluated as antagonists against *Sclerotium rolfsii*, with the fungicide Provax-200 serving as a positive control, and a culture of *S. rolfsii* without antagonists or fungicide as the negative control. In culture filtrate assays, dilutions of 10%, 20%, and 30% of the original concentration were tested. Inhibition of *S. rolfsii* mycelial growth in dual culture was determined by comparing mycelial growth in treated plates to that in control (untreated) plates. Values represent the mean±standard error (SE) for each treatment (*n*=3). Different letters within each column indicate significant differences among treatments, based on Fisher's LSD test (*p*<0.05).

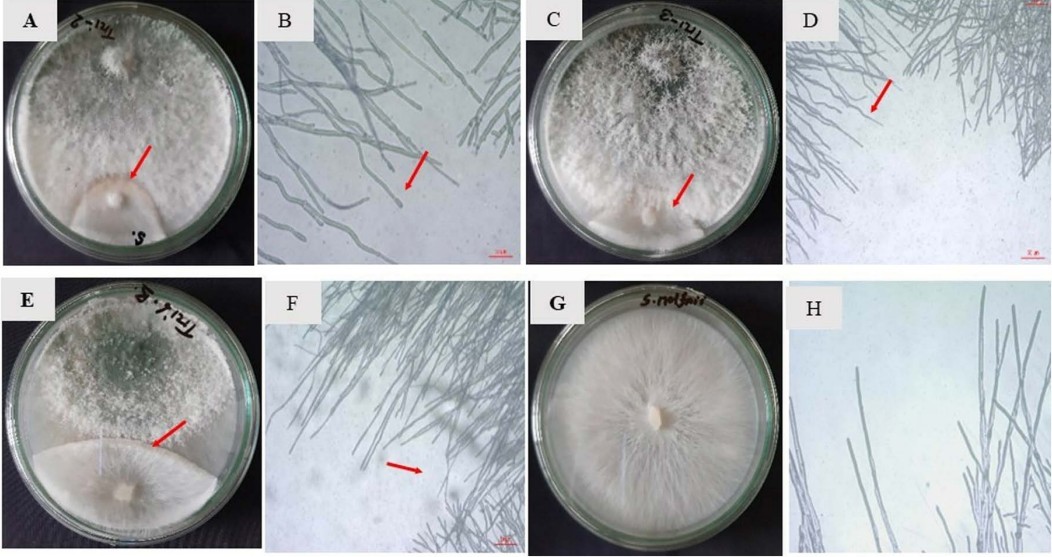

**Fig 1. *In vitro* interaction between *Trichoderma* isolates and *Sclerotium rolfsii* 5 days after inoculation (DAI) in a dual culture bioassay.** The interaction zone between *Trichoderma* and *S. rolfsii* was observed microscopically using a computer-connected microscope. Panels show different pairings: (A) *Trichoderma* isolate Tri2 + *S. rolfsii*, (C) *Trichoderma* isolate Tri3 + *S. rolfsii*, (E) *Trichoderma* isolate Tri6 + *S. rolfsii*, and (G) *S. rolfsii* (control). Red arrows in culture plates indicate the inhibition zone between *Trichoderma* and *S. rolfsii*. Microscopic observations revealed that *S. rolfsii* hyphae in the vicinity of *Trichoderma* appeared curved and wavy as indicated by red arrows (B, D, F), whereas those in the control plates exhibited a thin, elongated, and straight morphology (H).

Tri6 (72.22%) (Table 3). The use of the chemical control Provax-200 resulted in a complete (100%) inhibition of *S. rolfsii* growth. Microscopic observation revealed differences in the hyphal structure of *S. rolfsii*. In the control plates, the hyphae were long, thin, and linear (Fig 1H). In contrast, when *S. rolfsii* grew near *Trichoderma* isolates *Tri2* and *Tri3*, the hyphae appeared curved and wavy (Figs 1B, 1D, 1F), suggesting a possible disruption caused by *Trichoderma*. These results indicate that *Trichoderma* isolates, especially Tri2, Tri3, and Tri6, are effective antagonists against *S. rolfsii*, with their ability to inhibit fungal growth being comparable or superior to chemical treatments.

### Culture filtrate assay

The culture filtrate assay evaluated the effects of cell-free filtrates from various *Trichoderma* isolates on the mycelial growth of *S. rolfsii* at three different concentrations: 10%, 20%, and 30%. The filtrates of most *Trichoderma* isolates significantly inhibited the growth of *S. rolfsii* compared to the control (S1 Fig). At the 10% concentration, the inhibition ranged from 18.00% to 55.19%; at 20%, it ranged from 18.66% to 66.11%; and at 30%, it ranged from 30.65% to 88.78%. Among the isolates tested, Tri2, Tri3, and Tri6 demonstrated superior inhibition of *S. rolfsii* mycelial growth at all three concentrations when compared to the other isolates. The chemical control, Provax-200, achieved complete (100%) inhibition of *S. rolfsii* growth. Based on the antagonistic activity observed in both the culture assays and the culture filtrates, *Trichoderma* isolates Tri2, Tri3, and Tri6 were selected for further investigation as potential biocontrol agents.

### Molecular identification of the most biocontrol-agent-effective *Trichoderma*

The ITS sequences (S2 Fig) of selected fungal isolates were sequenced and checked for homology. Based on the Homology test of sequence data, the possible fungus species are listed in S2 Table. The accessions were recorded as OR678071, OR678072, and OR678073 in NCBI for Tri2, Tri3, and Tri6 isolates, respectively.

### Characterization of selected elite *Trichoderma* isolates for plant-growth-promoting and biocontrol traits

**Determination of phosphate solubilization activity.** The isolates Tri2 and Tri3 exhibited strong phosphate solubilization activity. On the other hand, Tri6 showed weak phosphate solubilization activity (S3 Table).

**Hydrolytic enzyme production.** The results in Supplementary S3 Table indicated that Tri2 and Tri3 exhibited strong cellulase activity, while Tri6 displayed moderate cellulase activity. A clear indication of cellulase production was the formation of a yellow halo around the colony after adding Congo red solution (S3A, S3B and S3C Figs). In contrast, Tri2, Tri3, and Tri6 demonstrated strong to moderate amylase and protease activities, respectively (S3 Table). The protease activity was confirmed by the color intensity observed after flooding with Bromocresol Green dye, indicating a protease reaction (S3D, S3E and S3F Figs; S3 Table). Similarly, amylase activity was confirmed by the color intensity after flooding with 1% iodine in 2% potassium iodide (S3G, S3H and S3I Figs; S3 Table). In the lipase test, isolates Tri2, Tri3, and Tri6 displayed strong to moderate lipase activity, with a clear white precipitation zone around the colony in the selective medium, indicating lipase reaction (S3J, S3K and S3L Figs; S3 Table). For the catalase test, Tri2 and Tri3 showed strong reactions, while Tri6 exhibited a moderate catalase reaction, confirming the presence of catalase (S3M, S3N and S3O Figs; S3 Table). Overall, Tri2 and Tri3 demonstrated strong to moderate activities across cellulase, amylase, protease, lipase, and catalase, while Tri6 showed moderate to weak enzyme activities.

**Inhibition of oxalic acid production by *S. rolfsii*.** The selected *Trichoderma* isolates (Tri2, Tri3, and Tri6) significantly reduced oxalic acid production by *S. rolfsii* in dual broth cultures (Table 4). In the absence of *Trichoderma*, *S. rolfsii* produced 2.98 mg of oxalic acid per milliliter of culture filtrate. However, in dual cultures with the *Trichoderma* antagonists, oxalic acid production was notably lower, ranging from 0.54 to 0.98 mg/mL. Among the three *Trichoderma* isolates, the lowest oxalic acid production (0.54 mg/mL) occurred in the dual culture with Tri2, followed by Tri3 (0.63 mg/mL). Given that Tri2 (81.87%), Tri3 (78.85%), and Tri6 (69.12%), along with their culture filtrates, exhibited the highest levels of oxalic acid degradation by *S. rolfsii*, these isolates were selected as elite antagonists for further studies against the pathogen (Table 4).

**Table 4. Inhibition of oxalic acid production of *Sclerotium rolfsii* by *Trichoderma* isolates Tri2, Tri3, and Tri6.**

| Isolates | Oxalic acid production by *S. rolfsii* (mg/mL) | (%) Decrease over control |
|---|---|---|
| *Sclerotium rolfsii* (control) | 2.98±0.12a | – |
| *Trichoderma* isolate Tri2+*S. rolfsii* | 0.54±0.12bc | 81.87 |
| *Trichoderma* isolate Tri3+*S. rolfsii* | 0.63±0.25bc | 78.85 |
| *Trichoderma* isolate Tri6+*S. rolfsii* | 0.92±0.03b | 69.12 |

Note: Values represent the mean±standard error (SE) for each treatment (*n*=3). Different letter(s) within each column indicate significant differences among treatments according to Fisher's LSD test (*p<0.05*).

### Effects of selected *Trichoderma* on the growth of tomato plant

***In vitro* assay.** The *in vitro* experiments revealed that treating seeds with *Trichoderma* significantly improved seed germination and seedling vigor compared to the untreated control (S4 Table). Seed germination rates varied between 71% to 94%, with the highest rate observed in seeds treated with Tri2 and the lowest in the control group. Germination rates increased by 13.14% to 33.34%, with the greatest improvement recorded in the seeds treated with Tri2 compared to the control. Seedling length also showed significant improvement with Tri2, Tri3, and Tri6, increasing by 100%, 75%, and 25%, respectively, compared to the control group. Correspondingly, vigor index values ranged from 283.67 to 756.67, with the highest vigor index recorded for seeds treated with Tri2 and the lowest for control seeds (S4 Table).

**Seed tray assay.** Tomato seeds treated with *Trichoderma*, whether in single, dual, or triple isolate combinations, showed significantly enhanced germination and seedling growth compared to the untreated control (S4 Fig). Germination was accelerated in treated seeds, with emergence occurring two days after sowing compared to four days in the control. After 10 days, the highest germination rate (96%) was observed in treatments T5 (Tri2+Tri3) and T6 (Tri2+Tri6), while the control (T1) showed the lowest (48%). Dual and triple combinations improved germination by 83–101%, whereas single treatments (Tri2, Tri3, Tri6) led to 16–38% increases over the control. (S4 Fig). *Trichoderma* also positively influenced seedling growth, such as shoot length, root length, the number of leaves, and whole seedling biomass in seed trays (Table 5). The maximum shoot length (14.87 cm) was observed in seedlings of T5, 168.68% higher than in controls. Root length (1.67 cm) was highest in T6, statistically similar to other consortium and most single treatments. Likewise, the number of leaves and seedling fresh biomass were significantly improved, particularly in dual and triple isolate treatments, with T5 showing the highest values (3.33 leaves/plant, 0.56 g biomass) (Table 5).

**Table 5. Effect of single, dual and triple *Trichoderma* treatment on seedling growth in seed trays.**

| Treatments | Shoot length (cm) | Root length (cm) | Plant fresh biomass (g) | No. of leaves/plant |
|---|---|---|---|---|
| T1 (Control) | 5.53±0.29d | 0.50±0.03c | 0.08±0.01e | 1.33±0.33c |
| T2 (Tri2) | 8.72±0.11c (57.71) | 1.50±0.28a (200.00) | 0.36±0.05 cd (341.43) | 2.67±0.33b (100.01) |
| T3 (Tri3) | 8.37±0.18c (51.21) | 1.47±0.05a (193.34) | 0.37±0.02 cd (345.57) | 2.00±0.33bc (50.00) |
| T4 (Tri6) | 7.03±0.48 cd (27.11) | 1.17±0.03b (133.34) | 0.25±0.04 cd (211.91) | 1.67±0.33c (25.01) |
| T5 (Tri2+Tri3) | 14.87±0.46a (168.88) | 1.63±0.09a (226.66) | 0.56±0.01a (580.44) | 3.33±0.33a (150.00) |
| T6 (Tri2+Tri6) | 11.50±0.17b (107.83) | 1.67±0.16a (233.34) | 0.45±0.01bc (442.77) | 3.00±0.33a (125.01) |
| T7 (Tri3+Tri6) | 11.50±0.76b (107.83) | 1.5±0.02a (200.00) | 0.46±0.01abc (454.92) | 3.00±0.33a (125.01) |
| T8 (Tri2+Tri3+Tri6) | 14.17±0.86ab (137.96) | 1.58±0.16a (216.66) | 0.59±0.06ab (507.33) | 3.30±0.33a (125.01) |

*Note*: In treatments, Tri2, Tri3, and Tri6 represent treatments with *Trichoderma* isolates Tri2, Tri3, and Tri6, respectively. Values (mean±SE) for each treatment were obtained from three biological replicates (*n*=3). Different letters within each column indicate significant differences, as determined by Fisher's LSD test (*p<0.05*). Values in parentheses represent the percentage increase relative to the control.

Photosynthetic segments such as chlorophyll *a*, chlorophyll *b*, total chlorophyll, and carotenoid content were significantly enhanced, particularly in consortium-treated seedlings compared to the control (S5 Table). The highest *Chla* (1.42) was noted in T6, followed by T5, T7, and T8. The maximum *Chlb* (0.82) was recorded from T5, statistically similar to T6, T7, and T8 treatments. Similarly, the highest total chlorophyll and carotenoid were observed in T5, showing a 322% and 222% increase compared to the control. The subsequent highest values were observed with T6, T7, and T8, indicating a statistically significant difference between the consortium and the single treatment (S5 Table).

**Potted plant assay.** Single, dual, and triple *Trichoderma* treatments significantly enhanced plant growth in potted tomato plants compared to non-inoculated controls (Table 6; Fig 2). The highest shoot length (30.93 cm) and fresh weight (31.37 g) were found in the T5 treatment (Tri2 + Tri3), which was statistically comparable to the T8 (Tri2 + Tri3 + Tri6) (29.83 cm and 31.15 g), T6 (Tri2 + Tri6) (28.67 cm and 29.90 g) and T7 (29.63 cm and 28.63 g). The maximum dry shoot weight (6.14 g) was recorded in T8 (Tri2 + Tri3 + Tri6), statistically similar to T5, T6, and T7. The maximum number of full-grown leaflets per plant (34.33) and stem (5.70 mm) and leaf diameter (7.42 mm) were found in plants treated with T5 (Tri2 + Tri3) (Table 6). The leaf number and stem and leaf diameters at T5 (Tri2 + Tri3) increased by 96.2%, 129% and 86.2%, respectively (Table 6). Root growth followed similar trends. T8 produced the longest roots (30.27 cm), while the highest root fresh and dry weights (7.98 g and 3.83 g) were found in T5 (Tri2 + Tri3), statistically similar to those in T8, T6, and T7 (Table 6).

There was a significant variation in SPAD value among the treatments between two different ages, 28 DAT and 42 DAT (S5 Fig). In the second week after transplant, the same value was observed among the single and consortium *Trichoderma* treatments but differed from the control. However, a significant difference was observed between single and consortium treatment during the 4th to 6th week of transplanting. Overall, six weeks after transplanting, the highest SPAD value was recorded from T8 (Tri2 + Tri3 + Tri6) (49.35), followed by T5 (49.12), T6 (47.52), and T7 (47.26), respectively (S5 Fig). The gas exchange parameters, including photosynthetic rate, leaf temperature, transpiration rate, and stomatal conductance to $H_2O$, were also positively influenced by single, dual, and triple *Trichoderma* treatment compared to the control (S6 Table). Compared with the control, the maximum photosynthetic rate and leaf temperature were recorded in T5, followed by T8, T6, and T7 (S6 Table). The T5 treatment resulted in a 307% and 390% increase in photosynthetic rate and leaf temperature over the nontreated controls. However, tomato plants inoculated with a consortium of *Trichoderma* such as T5, T6, T7, and T8 reduced transpiration rate significantly by −81%, −75%, −74%, and −75%%, respectively, and stomata conductance by −86%, −77%, −76% and −69%, respectively compared to the control plants (S6 Table).

The root colonization assays showed that *Trichoderma* successfully colonized the roots of tomato plants. The *Trichoderma* population in tomato roots increased with plant age (S6 Fig). Among three *Trichoderma* isolates, Tri2 showed the highest root population density ($2.67–18.36 \times 10^4$ cfu/g root fresh weight), followed by Tri3 ($2.00–15.01 \times 10^4$ cfu/g root fresh weight) and Tri6 ($1.33–13.16 \times 10^4$ cfu/g root fresh weight) (S6 Fig). Overall, six weeks after transplant, the maximum population densities ($27.95 \times 10^4$ cfu/g root fresh weight) were recorded in T8 (Tri2 + Tri3 + Tri6), followed by T5 ($27.78 \times 10^4$ cfu/g root fresh weight), T6 ($25.47 \times 10^4$ cfu/g root fresh weight) and T7 ($24.76 \times 10^4$ cfu/g root fresh weight) respectively (S6 Fig). These results show significant differences in *Trichoderma* root population densities observed between single and consortium *Trichoderma*-treated plants (S6 Fig).

## Suppression of damping-off disease by *Trichoderma* isolates in seed trays

The development of damping-off was rapid in unprotected tomato seedlings (T1) inoculated with only *S. rolfsii* compared to *Trichoderma*-treated seedlings (Fig 3). Damping-off symptoms were first observed in unprotected tomato seedlings (T1) one-week post-sowing. At the end of the assays, about 76% of the unprotected control seedlings had been infected by damping-off (Fig 3F). On the other hand, soil treated with single, dual, and triple *Trichoderma isolates* treatments resulted in significant control of the pathogen, causing a delay in symptom development and a lower number of seedlings with damping-off symptoms. In *Trichoderma*-treated seedlings, damping-off symptoms were observed in the fourth week

**Table 6. Effect of single, dual and triple combinations of *Trichoderma* treatment on growth promoting traits of tomato plants in pot treatment.**

| Treatments | Shoot | | | Stem and leaf | | | Root | | |
|---|---|---|---|---|---|---|---|---|---|
| | Length (cm) | Fresh weight (g) | Dry weight (g) | Full grown leaflet | Stem diameter (mm) | Leaf diameter (cm²/leaf area) | Length (cm) | Fresh weight (g) | Dry weight (g) |
| T1 (Control) | 19.33±0.17d | 14.32±0.02c | 1.04±0.02c | 17.50±0.29c | 2.49±0.29e | 3.98±0.36d | 20.35±0.18d | 2.97±0.43c | 1.37±0.09c |
| T2 (Tri2) | 22.50±0.29c (16.40) | 15.82±0.30c (10.50) | 2.60±0.14b (148.89) | 23.00±0.58b (31.43) | 3.85±0.10c (54.62) | 5.20±0.10c (30.55) | 23.67±0.67bc (16.28) | 4.68±0.27b (57.69) | 2.79±0.04b (103.90) |
| T3 (Tri3) | 21.50±0.29 cd (11.23) | 14.60±0.68c (1.93) | 2.07±0.12bc (98.41) | 22.00±0.58b (25.71) | 3.62±0.18 cd (45.51) | 5.13±0.07c (28.87) | 24.00±0.58c (17.92) | 4.56±0.15b (53.54) | 2.47±0.10b (80.49) |
| T4 (Tri6) | 20.67±0.60d (6.92) | 14.61±0.63c (2.00) | 2.22±0.50bc (113.10) | 21.67±0.88b (23.81) | 3.30±0.03d (32.40) | 5.13±0.07c (28.87) | 23.67±0.33c (16.28) | 4.56±0.07b (43.65) | 2.43±0.21b (78.04) |
| T5 (Tri2 + Tri3) | 29.83±0.44ab (54.34) | 31.37±0.67a (119.06) | 6.14±0.34a (488.83) | 34.33±1.20a (96.19) | 5.70±0.15a (129.05) | 7.42±0.19a (86.19) | 29.63±0.47ab (45.60) | 7.98±0.29a (168.80) | 3.83±0.14a (157.06) |
| T6 (Tri2 + Tri6) | 28.67±0.17b (48.30) | 29.90±0.30ab (108.78) | 5.27±0.32a (405.13) | 31.67±2.34a (80.95) | 5.62±0.17ab (125.70) | 7.07±0.30ab (77.41) | 28.90±0.67a (41.99) | 7.74±0.61a (160.49) | 3.35±0.07a (144.87) |
| T7 (Tri3 + Tri6) | 29.63±0.47ab (53.30) | 28.63±0.33b (99.95) | 5.07±0.33a (386.27) | 31.67±2.19a (80.95) | 5.17±0.02b (107.50) | 6.67±0.09b (67.37) | 29.20±0.89a (43.47) | 7.57±0.48a (154.88) | 3.51±0.04a (180.24) |
| T8 (Tri2 + Tri3 + Tri6) | 30.93±0.81a (60.03) | 31.15±1.09a (117.53) | 5.14±0.82a (392.67) | 34.00±1.00a (94.29) | 5.45±0.11ab (118.88) | 6.97±0.15ab (74.90) | 30.27±0.39a (48.71) | 7.80±0.76a (162.74) | 3.55±0.03a (159.51) |

*Note*: In treatments, Tri2, Tri3, and Tri6 represent treatments with *Trichoderma* isolates Tri2, Tri3, and Tri6, respectively. Values (mean±SE) for each treatment were obtained from three biological replicates ($n=3$). Different letters within each column indicate significant differences, as determined by Fisher's LSD test ($p < 0.05$). Values in parentheses represent the percentage increase relative to the control.

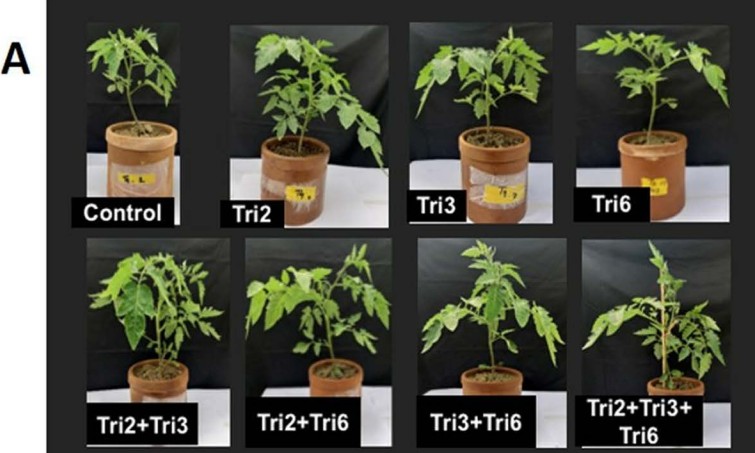

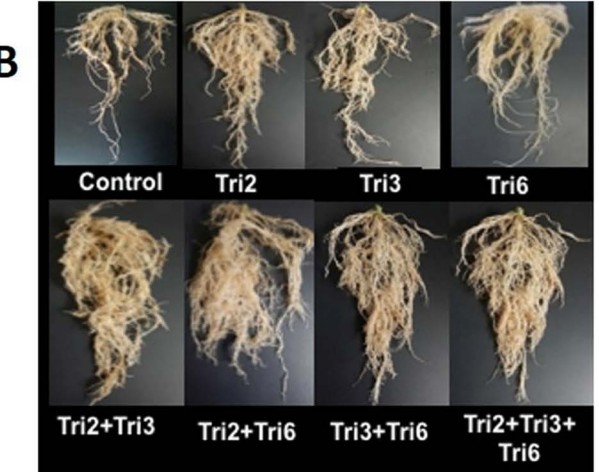

**Fig 2. Effect of single, dual, and triple combinations of *Trichoderma* treatment on the tomato plant: (A) shoot and (B) roots in the pot.** Sterilized soil mixed with autoclaved wheat grain was used as the control. In treatments, Tri2, Tri3, and Tri6 indicate treatment with wheat grain inoculum of *Trichoderma* isolate Tri2, Tri3, and Tri6, respectively.

post-sowing, and at the end of the experiment, only 8–13% of seedlings showed damping-off symptoms (Fig 3F). The percent reductions in damping-off infection caused by treatment with T8 *(*Tri2 + Tri3 + Tri6), T5 (Tri2 + Tri3), T6 (Tri2 + Tri6) and T7 (Tri3 + Tri6) over unprotected controls were 92%, 89%, 89%, and 87%, respectively. The treatments T2, T3, and T6 were statistically identical but differed from consortium treatments.

The number of sclerotia formed on the soil surfaces ranged from 21.00 to 0.83 per cm² across the various treatments, with the highest sclerotia formation observed in the unprotected control (Fig 3G). In contrast, *Trichoderma*-treated seed trays showed a marked reduction in the number of *S. rolfsii* sclerotia compared to the unprotected controls (21.00/cm²). Among the single and consortium treatments, the fewest sclerotia were found in T8 (1.00/cm² soil surface), followed by T5, T6, T7, T2, T3, and T4 by the end of the experiment. Additionally, damping-off and sclerotia formation occurred at the lowest rate in plants treated with Provax-200. No damping-off symptoms appeared in Provax-200-treated seedlings until the fourth week post-sowing, at which point only 2.47% of the seedlings exhibited symptoms. The lowest number of sclerotia (0.85/cm²) was observed on the soil surfaces in Provax-200-treated seed trays (Fig 3G).

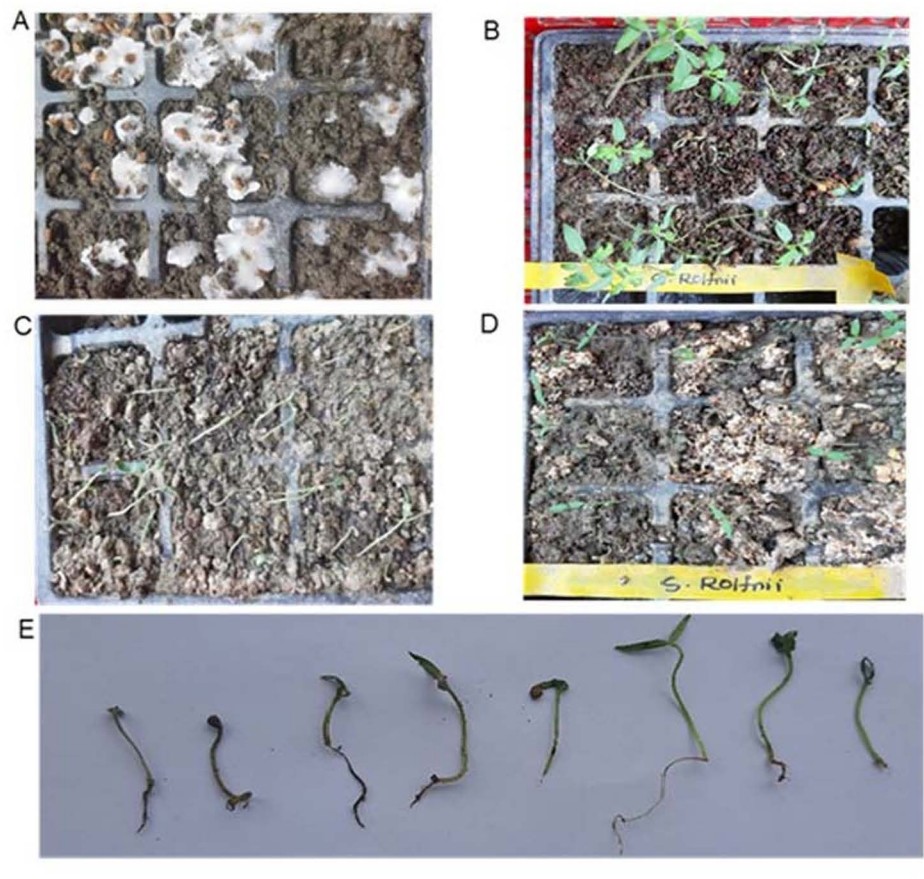

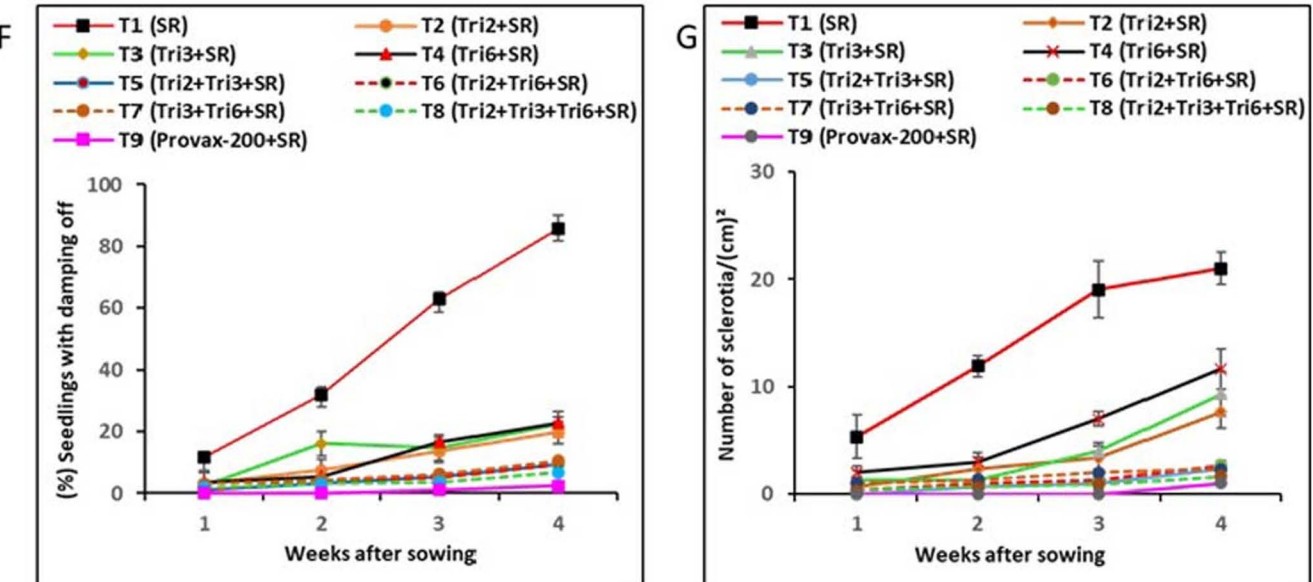

**Fig 3. Effect of application of single, dual, and triple combinations of *Trichoderma* on damping-off caused by *Sclerotium* rolfsii in tomato seedlings in a seed tray.** (A) white mycelia of *S. rolfsii* above the soil surface; (B-C) damping off of seedling in seed tray (D) presence of Sclerotia on soil surface in seed tray; (E) infected seedling; (F) percentage of seedlings with damping-off symptoms and (G) abundance of Sclerotia of *S. rolfsii* in the soils. Values represent mean±standard error (SE) for each treatment ($n = 3$). In the treatments, SR denotes inoculation with the Southern blight pathogen *Sclerotium rolfsii*, while Tri2, Tri3, and Tri6 represent treatments with *Trichoderma* isolates Tri2, Tri3, and Tri6, respectively. In T9, treatment with the fungicide Provax-200 was included.

## Suppression of southern blight disease by *Trichoderma* isolates in potted plants

Control tomato plants (unprotected) inoculated with *S. rolfsii* showed the first development of southern blight disease, which appeared in the third week after transplanting (Fig 4) and increased progressively with time. Most plants were severely affected by *S. rolfsii*, and the disease index was recorded to be 4.5 on a 0–5 rating scale at six weeks post-transplantation. All the *Trichoderma*-treated plants challenged with *S. rolfsii* showed a significant reduction in plant mortality compared with the control (Fig 4F). The maximum disease reduction (80%) was observed in plants treated with T8 (Tri2 + Tri3 + Tri6) and T5 (Tri2 + Tri3), followed by T6 (Tri2 + Tri3) (73%) and T7 (Tri2 + Tri6) (66%), respectively, in comparison to control (unprotected) plants. The inoculated plants treated with Provax-200 showed fewer disease symptoms and suffered no mortality. The fungicide-protected plants appeared healthy throughout the experiment.

## Biochemical changes in tomato plant during disease infection

**Hydrogen peroxide and malondialdehyde contents.** The levels of $H_2O_2$ and MDA were significantly higher in *S. rolfsii*-inoculated unprotected control plants (T1), with values of 1.83 nmol $g^{-1}$ FW and 11.57 µmol $g^{-1}$ FW, respectively,

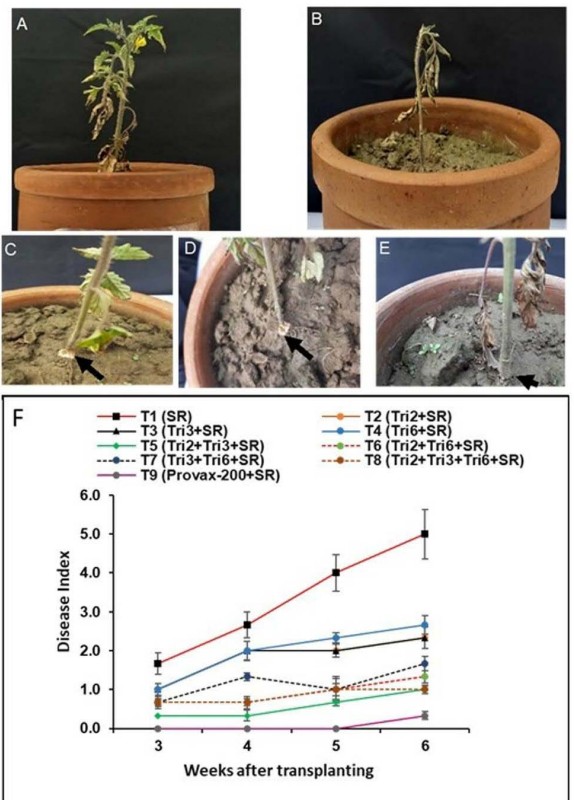

**Fig 4. Effect of single, dual, and triple combinations of *Trichoderma* isolates on Southern blight disease in tomato plants caused by *Sclerotium rolfsii* in a pot experiment.** (A) Initial discoloration of lower leaves; (B-C) onset of wilting in potted plants; (D) whitish mycelial growth on the lower stem just above the soil line; (E) stem girdling near the soil line. (F) Disease progression post-transplantation, as assessed by the mean disease score (*n* = 9). Disease scale (0-5): 0 = no symptoms; 1 = few leaves wilting; 2 = slight infection with mycelial mat on soil surface only; 3 = moderate infection, showing wilting and blighting with mycelial mat on stem base; 4 = severe infection with advanced wilting and abundant sclerotia at the stem base; 5 = plant death. Values represent the mean ± standard error (SE) for each treatment. In the treatments, SR denotes inoculation with the Southern blight pathogen *S. rolfsii*, while Tri2, Tri3, and Tri6 represent treatments with *Trichoderma* isolates Tri2, Tri3, and Tri6, respectively. In T9, treatment with the fungicide Provax-200 was included.

compared to plants treated with single, dual, and triple *Trichoderma* isolates (Figs 5A and 5B). The lowest $H_2O_2$ and MDA levels were observed in *S. rolfsii*-inoculated plants treated with T5 (Tri2 + Tri3), with values of 0.29 nmol g$^{-1}$ FW and 4.49 µmol g$^{-1}$ FW. Statistically similar results were found in plants treated with T8 (Tri2 + Tri3 + Tri6) (0.30 nmol g$^{-1}$ FW and 4.55 µmol g$^{-1}$ FW), T6 (Tri2 + Tri6) (0.33 nmol g$^{-1}$ FW and 4.65 µmol g$^{-1}$ FW), and T7 (0.38 nmol g$^{-1}$ FW and 4.65 µmol g$^{-1}$ FW), respectively. In contrast, plants treated with single *Trichoderma* isolates (T1, T2, and T3) exhibited significantly higher levels of $H_2O_2$ and MDA than those treated with dual or triple *Trichoderma* isolates (Figs 5A and 5B).

**Phenolic and flavonoid contents.** Phenolics and flavonoids are non-enzymatic antioxidants produced by plants as part of their basal defense response. A significant increase in phenolic and flavonoid contents was observed in all *Trichoderma*-treated plants inoculated with *S. rolfsii*, compared to the plants inoculated with *S. rolfsii* alone (unprotected control, T1) (Figs 5C and 5D). The phenolic and flavonoid contents ranged from 0.48 to 1.94 µg g$^{-1}$ FW and 0.99 to 2.20 µg g$^{-1}$ FW, respectively. Treatment with T5 (Tri2 + Tri3) resulted in the highest phenolic and flavonoid contents, showing a 306% and 122% increase, respectively, compared to the control. The second-highest content was observed in T8 (Tri2 + Tri3 + Tri6), followed by T6 (Tri2 + Tri6) and T7, with these treatments being statistically similar. Plants treated with single *Trichoderma* isolates (T2, T3, and T4) also exhibited significantly higher phenolic and flavonoid contents than the pathogen-inoculated plants, although they differed significantly from the consortium treatments (Figs 5C and 5D).

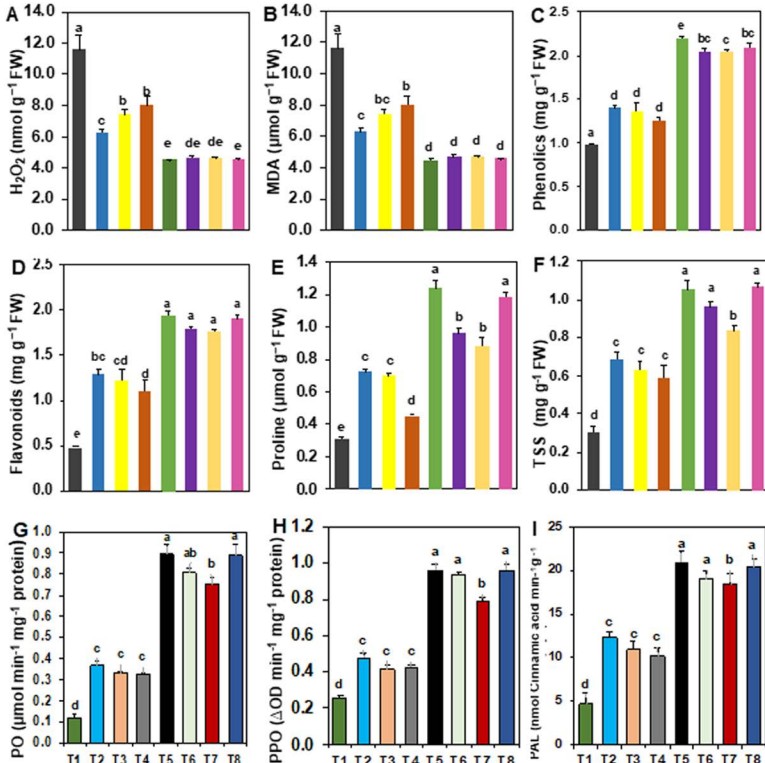

**Fig 5. Effect of single, dual, and triple combinations of *Trichoderma* isolates on biochemical parameters in tomato plants inoculated with Southern blight (*Sclerotium rolfsii*) in pot experiments.** (A) Hydrogen peroxide ($H_2O_2$) content; (B) malondialdehyde (MDA) content; (C) total phenolic content; (D) total flavonoid content; (E) proline content; (F) total soluble sugar (TSS) content; (G) peroxidase (PO) activity; (H) polyphenol oxidase (PPO) activity; and (I) phenylalanine ammonia-lyase (PAL) activity. Values are presented as means ± SE, with different letters above the bars indicating statistically significant differences ($p < 0.05$, LSD test). Treatments include: T1 = *S. rolfsii* alone; T2 = *Trichoderma* isolate Tri2 + *S. rolfsii*; T3 = *Trichoderma* isolate Tri3 + *S. rolfsii*; T4 = *Trichoderma* isolate Tri6 + *S. rolfsii*; T5 = *Trichoderma* isolates Tri2 + Tri3 + *S. rolfsii*; T6 = *Trichoderma* isolates Tri2 + Tri6 + *S. rolfsii*; T7 = *Trichoderma* isolates Tri3 + Tri6 + *S. rolfsii*; and T8 = *Trichoderma* isolates Tri2 + Tri3 + Tri6 + *S. rolfsii*.

**Soluble sugar and proline contents.** Proline and soluble sugar levels were significantly decreased in plants infected with *S. rolfsii* (unprotected control, T1), and this reduction was reversed in plants treated with *Trichoderma* inoculation (Figs 5E and 5F). Among all the treatments, the lowest levels of proline and soluble sugar were observed in pathogen-inoculated plants. In contrast, the highest increases in proline (251%) and soluble sugar (301%) were recorded in T8 (Tri2 + Tri3 + Tri4), which showed statistically similar results to T5, T6, and T7. Plants treated with single *Trichoderma* isolates also exhibited significant increases in proline and soluble sugar compared to the control (T1) (Figs 5E and 5F).

## Peroxidase (PO), phenylalanine ammonia-lyase (PAL), and polyphenol oxidase (PPO) content

The activity of PO, PPO, and PAL was significantly higher in all *Trichoderma*-treated plants (single, dual, and triple isolates) than in the *S. rolfsii* inoculated unprotected control plants (T1) (Figs 5G, 5H and 5I). PO activity was enhanced in plants treated with *Trichoderma* and inoculated with the pathogen. The highest PO activity was recorded in plants treated with T5 (Tri2 + Tri3 + *S. rolfsii*) and T8 (Tri2 + Tri3 + Tri4 + *S. rolfsii*), both showing 0.89 µmol min$^{-1}$ mg$^{-1}$ protein, followed by T6 (Tri2 + Tri6) at 0.80 µmol min$^{-1}$ mg$^{-1}$ protein, T7 at 0.87 µmol min$^{-1}$ mg$^{-1}$ protein, and T2, T3, and T4 at 0.37 and 0.33 µmol min$^{-1}$ mg$^{-1}$ protein, respectively. Similarly, the highest PPO activity was recorded in T5 (Tri2 + Tri3 + *S. rolfsii*), T6, and T8 (Tri2 + Tri3 + Tri6 + *S. rolfsii*), followed by T7. Furthermore, significant increases in PPO activity were observed in Tri2, Tri3, and Tri6-treated plants, with increases of 89%, 64%, and 63%, respectively (Fig 5H). The highest PAL activity (20.89 nmol cinnamic acid min$^{-1}$ g$^{-1}$ protein) was recorded in T5 (Tri2 + Tri3 + *S. rolfsii*)-treated plants, followed by T8 (20.36 nmol cinnamic acid min$^{-1}$ g$^{-1}$ protein), T6 (18.96 nmol cinnamic acid min$^{-1}$ g$^{-1}$ protein), and T7 (18.43 nmol cinnamic acid min$^{-1}$ g$^{-1}$ protein) (Fig 5I). These results demonstrate that consortium treatments induced significantly higher PO, PAL, and PPO activities compared to single treatments.

## Suppression of southern blight and yield of tomato in the field

Field data revealed that southern blight disease developed more rapidly in plants grown in pathogen-inoculated plots compared to those in *Trichoderma*-treated plots. Disease symptoms appeared within three weeks of pathogen inoculation in the control (unprotected) plants and progressively increased over time (Fig 6). In contrast, the protected plants remained nearly healthy throughout the experiment, with reduced mortality and a disease index ranging from 3.00 to 1.33 in plants treated with *Trichoderma* at nine weeks post-inoculation. The highest disease reduction was observed in plants treated with the consortium of Tri2 + Tri3 + Tri6 (T8) and Tri2 + Tri3 (T5), both showing a 73% reduction, followed by Tri2 + Tri3 (T6) with a 66% reduction, and Tri2 + Tri6 (T7) with a 65% reduction, all compared to the unprotected control plants. Additionally, plants treated with single *Trichoderma* isolates also exhibited a significant decrease in disease incidence compared to the *S. rolfsii*-inoculated controls. Inoculated plants treated with Provax-200 showed fewer disease symptoms and no mortality, remaining healthy throughout the experiment (Fig 6D).

The SPAD values were significantly influenced by the application of *Trichoderma* isolates, either singly, in dual or triple combinations, as well as by the fungicide Provax-200, in tomato plants challenged with *S. rolfsii* under field conditions (S7 Fig). Across all time points (3, 5, 7, and 9 weeks after transplanting), the untreated pathogen-inoculated control (T1) exhibited the lowest SPAD values, indicating severe chlorophyll degradation and stress induced by the pathogen. Among the treatments, the triple combination T8 (Tri2 + Tri3 + Tri6 + SR) and the dual combinations T5 (Tri2 + Tri3 + SR) and T6 (Tri2 + Tri6 + SR) consistently exhibited the highest SPAD values throughout the study. Notably, T8 recorded the maximum SPAD value at 7 weeks (53.63), closely followed by T5 (53.52) and T6 (52.97). At 9 weeks, T5 maintained the highest SPAD value (47.80), suggesting sustained chlorophyll content and photosynthetic efficiency under pathogen stress.

The results also demonstrated that inoculating tomato plants with *Trichoderma* isolates, whether singly, in pairs, or as a trio, significantly increased plant height, yield parameters, and fruit brix content compared to the unprotected pathogen-inoculated control plants in the field (Table 7). Plants inoculated with Tri2 + Tri3 *(*T5) showed a maximum increase (94%) in plant height as compared to other treatments. Similarly, the maximum fruit number was increased by 114% with the

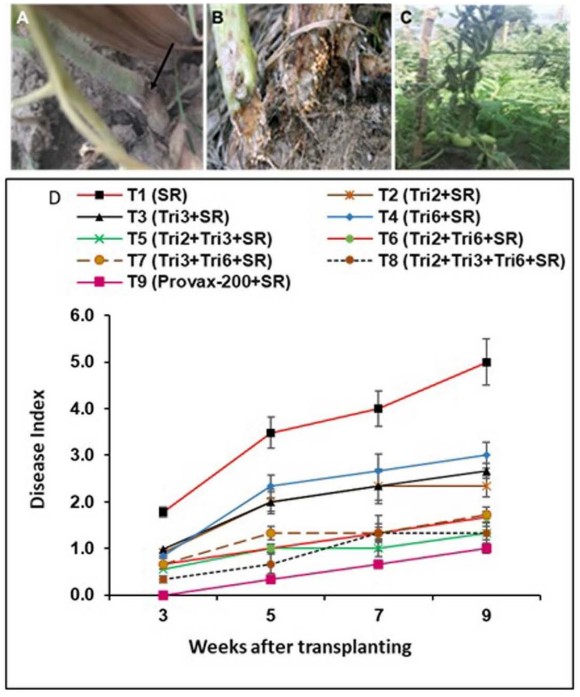

**Fig 6. Effect of single, dual, and triple combinations of *Trichoderma* isolates on Southern blight disease in tomato plants caused by *Sclerotium rolfsii* in field experiments.** (A) Lesion formation on the stem; (B) presence of sclerotia on the stem base; (C) wilting observed in tomato plants; (D) Disease progression post-transplantation, assessed by the mean disease scale (n = 9). Disease scale (0-5): 0 = no symptoms; 1 = few leaves wilting; 2 = slight infection with mycelial mat on soil surface only; 3 = moderate infection, showing wilting and blighting with mycelial mat on stem base; 4 = severe infection with advanced wilting and abundant sclerotia at the stem base; 5 = plant death. Values represent the mean ± standard error (SE) for each treatment. In the treatments, SR denotes inoculation with the Southern blight pathogen *Sclerotium rolfsii*, while Tri2, Tri3, and Tri6 represent treatments with *Trichoderma* isolates Tri2, Tri3, and Tri6, respectively. In T9, treatment with the fungicide Provax-200 was included.

same microbial combinations (Tri2 + Tri3), which showed statistically the same increment with treatments T8, T6, and T7, respectively. The yield of tomato was significantly increased due to single and consortium application of *Trichoderma* isolates compared to the unprotected control. The yield of tomatoes ranged from 6.9 to 19.00 t/ha in various treatments in the field. Among the treatments, soil amendment with Tri2 + Tri3 (T5) gave the highest yield (19.59 t/ha$^{-1}$) and Brix content (4.88), which was statistically similar to T8, T6, and T7 treatments. The lowest yield and Brix contents (6.9 t/ha$^{-1}$ and 2.4) were recorded from control plants (Table 7). These fungicide-protected plants were also significantly taller and gave substantially higher fruit yields than the inoculated control plants.

## Discussion

Several studies have established the positive function of *Trichoderma* isolates in the biological control of *S. rolfsii* in various crops. Application of *Trichoderma*-fortified inoculum in the soil was found to be very effective in reducing the disease caused by *S. rolfsii* in potatoes [33]; carrot [34]; soybean [32] and tomato [58]. Nevertheless, these studies focused indistinctly on the potential disease suppression mechanisms of *Trichoderma* isolates.

The present study evaluated the growth promotion and biocontrol efficacy of three indigenous *Trichoderma* strains against the southern blight pathogen *S. rolfsii*. It elucidated the biochemical mechanisms involved in the *Trichoderma*-mediated disease suppression in tomato plants. The cultures and culture filtrates of various *Trichoderma* species have been narrated to show antagonism towards *S. rolfsii* [22,59,60]. However, the degree of antagonism by different *Trichoderma* isolates against *S. rolfsii* often varies in dual culture tests [61]. The antagonism conferred by *Trichoderma* isolates

**Table 7. Effect of *Trichoderma* application on yield-contributing parameters in *Sclerotium rolfsii*-inoculated tomato plants in field conditions.**

| Treatment | Plant height (cm) | Fruit number/plot | Yield (ton/ha) | Brix content |
|---|---|---|---|---|
| T1 (SR) | 40.33±0.55g | 41.78±2.30f | 6.97±0.26d | 2.43±0.28d |
| T2 (Tri2+SR) | 65.48±0.30de (62.34) | 65.78±1.68d (57.44) | 13.16±0.72c (88.85) | 3.31±0.04c (36.03) |
| T3 (Tri3+SR) | 61.78±0.11ef (53.17) | 63.22±2.39de (51.33) | 12.76±0.39c (83.25) | 3.28±0.01c (34.80) |
| T4 (Tri6+SR) | 58.67±1.00f (45.48) | 57.83±0.48e (38.43) | 12.62±0.31c (81.24) | 3.18±0.04c (30.69) |
| T5 (Tri2+Tri3+SR) | 78.37±0.27a (94.31) | 89.43±2.08a (114.08) | 19.59±0.05a (181.18) | 4.88±0.04a (100.42) |
| T6 (Tri2+Tri6+SR) | 72.13±0.09bc (78.85) | 76.44±4.16bc (82.97) | 18.49±0.29ab (165.39) | 4.49±0.01b (84.39) |
| T7 (Tri3+Tri6+SR) | 68.63±0.32 cd (70.16) | 72.89±1.42c (74.47) | 17.49±0.50b (150.39) | 4.47±0.15b (85.57) |
| T8 (Tri2+Tri3+Tri6+SR) | 73.85±2.91ab (83.10) | 80.00±4.16b (91.49) | 19.00±0.13a (172.71) | 4.67±0.04ab (92.06) |
| T9 (Provax-200+SR) | 73.29±3.35bc (81.72) | 78.22±1.42bc (87.23) | 18.76±0.73ab (169.37) | 4.50±0.08b (84.80) |

*Note*: In the treatments, SR denotes inoculation with the Southern blight pathogen *Sclerotium rolfsii*, while Tri2, Tri3, and Tri6 represent treatments with *Trichoderma* isolates Tri2, Tri3, and Tri6, respectively. In T9, treatment with the fungicide Provax-200 was included. Values (mean±SE) for each treatment were obtained from three biological replicates ($n=3$). Different letters within each column indicate significant differences, as determined by Fisher's LSD test ($p<0.05$).

also varies with different plant pathogenic fungi. Some isolates of *Trichoderma* have been reported to act by antibiosis and parasitism on various fungal pathogens. Antibiosis and competition for nutrients and energy are other prominent mechanisms of *Trichoderma* against pathogens. However, the degree of antibiosis against *S. rolfsii* can vary among *Trichoderma* isolates [62].

The *Trichoderma* isolates in the present study showed considerable efficacy in promoting tomato plant growth, including seed germination and seedling vigor. For biological inoculants, germination and vigor are regarded as essential selection features since they provide the plant with an early advantage that may be beneficial for its future growth. Moreover, introducing the selected *Trichoderma* into the tomato root systems successfully augmented the growth and photosynthesis apparatus of the plants. The selected *Trichoderma* also proved to be effective and competent biocontrol agents for protecting seedlings and field-grown tomato plants against the deadly pathogen *S. rotfsii*. Application of these strains, both in single and consortium, significantly enhanced the growth, and yield of tomatoes.

Increased amylase activity and phosphate solubilization capability of these strains were observed when tested *in vitro*. The amylase enzyme breaks down starch into glucose, an energy source for the enhanced germination process [63]. The enhanced germination rate of tomato plants when treated with the selected *Trichoderma* isolates might be a result of the capability of the isolates to excrete amylase enzymes. Furthermore, enhanced growth parameters, such as plant shoot and root length, stem and leaf diameter, shoot weight, root weight, and number of full-grown leaflets, resulted from the single and combined application of *Trichoderma* isolates may be attributed to their fortified capacity of nutrient mobilization, such as phosphate solubilization [22]. This result resembles previous reports where *Trichoderma* induced phosphate mobilization and increased the quantity of phosphorus in plants, resulting in enhanced growth parameters [64–65]. However, the degree of seedling growth and vigor depended on the *Trichoderma* applied either in a single or in a combined form, which is similar to the findings of earlier studies where the variation of isolates resulted in different levels of seedling growth and vigor [66]. The increased amount of photosynthetic pigments indicates enhanced formation of storage materials, which is attributed to the augmentation of plant growth. While carotenoids not only act as attractants for pollinators and light-harvesting pigments, they also act as an antioxidant that has a protective role against pathogen-induced oxidative stress [67–68]. Applying *Trichoderma isolates* improved photosynthetic pigments and carotenoids, which might have supported the enhanced growth parameters and improved disease suppression capacity against *S. rolfsii*. Reduction of transpiration rate and stomatal conductance to $H_2O$ can be mediated by the antibiotics produced by *Trichoderma* [69]. Based on the results obtained from the current experiment, it is reasonable to suggest that the use of *Trichoderma*

enhances photosynthetic productivity by increasing photosynthetic pigments. Additionally, it improves leaf temperature by reducing stomatal aperture, thus minimizing water loss through transpiration and preventing harmful ions from entering the transpiration stream. Root colonization assay presented a noteworthy increase in the *Trichoderma* population after transplanting the plant into the pot. This result is similar to a previous study where increased *Trichoderma* colonization was observed [70]. The competition and antagonistic activities of *Trichoderma* against other inhabitants in the rhizosphere might have resulted in this increased root colonization.

During mycoparasitic interaction, cellulase degrades the cell wall of phytopathogenic fungi. It also contributes to the breaching of host-pathogen cell walls at positions of intended penetration of *Trichoderma* ( [71]. Protease also degrades the pathogen cell wall and releases protein by lysis, which later acts as a nutrient source for the mycoparasites [72]. *Trichoderma* demonstrated aggressive biocontrol ability towards *S. rolfsii* by liberating extracellular hydrolytic enzymes such as protease, pectinase, amylase, and cellulase activity [73]. Therefore, in this study, positive responses of *Trichoderma* isolates for cellulase, lipase, protease, amylase, and catalase production might have contributed to the reduced growth of *S. rolfsii* and disease severity. Oxalic acid is responsible for degrading the host tissue by *S. rolfsii*; therefore, it is a crucial pathogenicity factor [74]. Reducing the amount of oxalic acid leads to the slower establishment of the pathogen, as degradation of plant tissue is prevented. *In vitro* tests of *Trichoderma* isolates showed that these strains significantly inhibited oxalic acid production by *S. rolfsii*, thereby resulting in the slow establishment of disease in the *Trichoderma*-treated plants. *Trichoderma* likely optimized the disease management by neutralizing this pathogenicity factor of *S. rolfsii*.

A previous study showed that *Trichoderma*-treated brinjal plants showed significantly higher growth and vigor than the untreated *Sclerotinia sclerotiorum* inoculated control. Maximum chlorophyll content, shoot length, and yield were recorded in consortium-treated seedlings than in individual *Trichoderma*-treated seedlings [29]. The root colonization capacity of *Trichoderma* was also found to be higher in combined application than in single treatment [75]. Similar results have been found in the current experiment, as the combined effect of *Trichoderma* isolates performed well compared with a single treatment in the case of all the growth parameters, extracellular enzyme performance, root colonization ability, and disease suppression ability.

The generation of excessive levels of $H_2O_2$ and MDA due to stress can lead to lipid peroxidation, which can adversely affect the cell membrane [76]. However, a recent study has shown that *Trichoderma*-protected tomato plants have significantly lower levels of $H_2O_2$ and MDA compared to unprotected plants. This suggests that *Trichoderma* may increase the expression of enzymes and the synthesis of compounds that help eliminate toxic molecules associated with lipid peroxidation, such as reactive oxygen species (ROS) [17,77]. This antioxidative mechanism is believed to be part of induced systemic resistance (ISR) [78]. As a result, defense enzymes such as PO, PAL, and PPO are upregulated, and the accumulation of osmoprotectants is stimulated, thereby fortifying the plant's tolerance against pathogen-induced oxidative damage.

Phenolic substances regulate signal molecules directly related to defense-related genes [79]. Studies have shown that plants treated with antagonist fungi have significantly higher levels of phenolic substances, which can induce a plant defense response by changing various metabolic processes [80]. Singh et al. [81] evaluated the phenolic and flavonoid contents, where different microbial combinations showed a significantly higher amount of phenolics and flavonoids than a single application. The present study implied an increased accumulation of phenol and flavonoid in a consortium of *Trichoderma* than that of a single application. However, both single and consortium applications of *Trichoderma* showed higher accumulations of phenol and flavonoid, indicating improved plant defense. Compared to unprotected tomato control plants, an increased amount of proline and soluble sugar was also observed in *Trichoderma*-treated plants. The ability of proline to affect peroxidase activity aids in the lignification of cell walls. This, in turn, enhances the plant's ability to resist pathogens [82]. During stress, soluble sugar enhances osmo-protection and releases plants from oxidative stress by scavenging ROS [83]. *Trichoderma*-treated tomato plants showed increased activities of PO, PAL, and PPO, which can be correlated with enhanced defense against southern blight infection. It has been reported that *Trichoderma*, both in single

and consortium applications, increases the defense enzyme activity, which contributes to the suppression of *Alternaria solani* [31]. In an earlier study, potato plants treated with *Trichoderma* and inoculated with *R. solani* showed significantly superior activity of PAL in consortium application than that of a single application [84].

It is apparent that the ability of *Trichoderma* isolates Tri2, Tri3, and Tri6 to suppress *S. rolfsii* and enhance plant growth can be attributed to multiple mechanistic layers. These strains likely secrete a suite of hydrolytic enzymes, including cellulases, proteases, and chitinases, that degrade fungal cell walls and suppress pathogen proliferation. Additionally, the observed reduction in oxalic acid levels suggests that the isolates interfere with *S. rolfsii* virulence by either enzymatically degrading oxalic acid or altering the rhizosphere pH to neutralize its effect. Importantly, *Trichoderma* is known to act as a microbe-associated molecular pattern (MAMP), which primes host plant immunity through systemic resistance pathways [85]. This priming can lead to upregulation of antioxidant enzymes such as PO, PPO, and PAL, as well as osmo-protectants (e.g., proline and flavonoids), collectively enhancing cellular detoxification and structural reinforcement under pathogen attack. Thus, the combined action of direct antagonism, virulence factor suppression, and host defense modulation may form the basis of the observed biocontrol efficacy. However, further research is necessary to clarify the precise mechanism involved in the biocontrol activity of *Trichoderma* isolates.

## Conclusions

The study showcases the significant potential of indigenous *Trichoderma* isolates in promoting robust growth and effectively suppressing *S. rolfsii*-induced diseases in tomatoes. The consortium application of these isolates demonstrates superior outcomes across various parameters compared to individual treatments. The findings underscore the multi-faceted mechanisms underlying *Trichoderma*-mediated plant defense, including extracellular enzyme production, inhibition of pathogenicity factors, and enhanced antioxidative responses. This research emphasizes the promising role of *Trichoderma* as a biocontrol strategy for sustainable tomato production.

## Supporting information

**S1 Fig. Antagonism between *Trichoderma* cell-free culture filtrate and *Scelrotium rolfsii* at different concentrations (10%, 20%, and 30%) at 7 days after inoculation (DAI).** Panels show the following treatments: Tri2 (A – 10%, B – 20%, C – 30%), Tri3 (D – 10%, E – 20%, F – 30%), Tri6 (G – 10%, H – 20%, I – 30%), J – Provax-200 at 200 ppm, and K – S. *rolfsii* (control).
(TIF)

**S2 Fig. ITS profiles generated using ITS4 and ITS5 primers from fungi.** M: 1 kb DNA ladder (marker).
(TIF)

**S3 Fig. Enzyme activity of three selected *Trichoderma* isolates in a plate assay at 5 days after inoculation (DAI).** Panels show the following enzyme reactions: cellulose (A – C), protease (D- F), amylase (G – I), lipase (J – L), and catalase (M – O). Tri2, Tri3, and Tri6 represent different *Trichoderma* isolates.
(TIF)

**S4 Fig. Effect of single, dual, and triple combinations of *Trichoderma* treatments on the germination percentage (mean ± SE) of tomato seedlings in the seed tray.** Tri2, Tri3, and Tri6 represent different *Trichoderma* isolates. A Fisher's LSD test (p < 0.05) was performed to identify significant differences among the treatments.
(TIF)

**S5 Fig. SPAD values of tomato plant leaves grown with single, dual, and triple combinations of *Trichoderma* treatments in pots.** SPAD readings were recorded at 4, 5, 7, and 9 weeks after transplanting. In the control treatment (T1), plants were not inoculated with any *Trichoderma* isolates. For the other treatments, Tri2, Tri3, and Tri6 represent

plants treated with *Trichoderma* isolates Tri2, Tri3, and Tri6, respectively. The data are presented as mean±SE, with values derived from three biological replicates (n=3) for each treatment.
(TIF)

**S6 Fig. Population of *Trichoderma* in the roots of tomato plants, measured under single, dual, and triple combinations of *Trichoderma* treatments at 2, 4, 5, and 6 weeks after transplanting.** The data are presented as the number of colony-forming units (CFU) per gram of fresh root weight (mean±standard error). Root samples were collected from three sets of four plants at each time point (2, 4, 5, and 6 weeks post-transplant). In the treatments, Tri2, Tri3, and Tri6 represent different *Trichoderma* isolates.
(TIF)

**S7 Fig. SPAD values of tomato plant leaves grown under different treatments and inoculated with *Sclerotium rolfsii* under field conditions.** SPAD readings were recorded for 4, 5, 7, and 9-week-old transplanted tomato plants across various treatments. In the treatments, "SR" indicates inoculation with the Southern blight pathogen *Sclerotium* rolfsii, while Tri2, Tri3, and Tri6 represent treatments with *Trichoderma* isolates Tri2, Tri3, and Tri6, respectively. Treatment T9 includes the application of the fungicide Provax-200. Data are presented as mean±SE, with values obtained from three biological replicates (n=3) for each treatment.
(TIF)

**S1 Table. List of *Trichoderma* isolates, host plants and locations of collections.**
(DOCX)

**S2 Table. Sequencing homology test of the ITS region of the selected *Trichoderma* isolates with significant alignments using BLAST software.**
(DOCX)

**S3 Table. Sequencing homology test of the ITS region of the selected *Trichoderma* isolates with significant alignments using BLAST software.** (+) Isolates showing weak enzyme activity, (++) Isolates showing moderate enzyme activity and (+++) Isolates showing strong enzyme activity.
(DOCX)

**S4 Table. Effect of selected *Trichoderma* isolates on the tomato seed germination, seedling height and seedling vigor in in vitro assays.** Values (mean±SE) for each treatment were obtained from three biological replicates (n=3). Different letters within each column indicate significant differences, as determined by Fisher's LSD test (p<0.05). Values in parentheses represent the percentage increase relative to the control.
(DOCX)

**S5 Table. Effect of single, dual and triple combinations of *Trichoderma* treatments on photosynthetic pigments in 21-day seedlings of tomato.** In treatments, Tri2, Tri3, and Tri6 represent treatments with *Trichoderma* isolates Tri2, Tri3, and Tri6, respectively. Values (mean±SE) for each treatment were obtained from three biological replicates (n=3). Different letters within each column indicate significant differences, as determined by Fisher's LSD test (p<0.05). Values in parentheses represent the percentage increase relative to the control.
(DOCX)

**S6 Table. Gas exchange attributes of tomato foliage in plants treated with single, dual, and triple combinations of *Trichoderma* isolates.** In treatments, Tri2, Tri3, and Tri6 represent treatments with *Trichoderma* isolates Tri2, Tri3, and Tri6, respectively. Values (mean±SE) for each treatment were obtained from three biological replicates (n=3). Different

letters within each column indicate significant differences, as determined by Fisher's LSD test (p < 0.05). Values in parentheses represent the percentage increase relative to the control.
(DOCX)

**S7 Table. Effect of application single, dual, and triple combinations of *Trichoderma* on damping-off caused by *Sclerotium rolfsii* in tomato seedlings in seed trays at different weeks after sowing (WAS).** In the treatments, SR denotes inoculation with the Southern blight pathogen *Sclerotium* rolfsii, while Tri2, Tri3, and Tri6 represent treatments with *Trichoderma* isolates Tri2, Tri3, and Tri6, respectively. In T9, treatment with the fungicide Provax-200 was included. Values (mean ± SE) for each treatment were obtained from three biological replicates (n = 3). Different letters within each column indicate significant differences, as determined by Fisher's LSD test (p < 0.05).
(DOCX)

**S8 Table. Effect of application of single, dual, and triple combinations of *Trichoderma* on the production of Sclerotia on the soil surface by *Sclerotium rolfsii* in seed trays at different weeks after sowing (WAS).** In the treatments, SR denotes inoculation with the Southern blight pathogen *Sclerotium* rolfsii, while Tri2, Tri3, and Tri6 represent treatments with *Trichoderma* isolates Tri2, Tri3, and Tri6, respectively. In T9, treatment with the fungicide Provax-200 was included. Values (mean ± SE) for each treatment were obtained from three biological replicates (n = 3). Different letters within each column indicate significant differences, as determined by Fisher's LSD test (p < 0.05).
(DOCX)

**S9 Table. Effect of single, dual, and triple combinations of *Trichoderma* isolates on Southern blight disease in tomato plants caused by *Sclerotium rolfsii* in a pot experiment at different weeks after transplanting (WAT).** In the treatments, SR denotes inoculation with the Southern blight pathogen *Sclerotium* rolfsii, while Tri2, Tri3, and Tri6 represent treatments with *Trichoderma* isolates Tri2, Tri3, and Tri6, respectively. In T9, treatment with the fungicide Provax-200 was included. Values (mean ± SE) for each treatment were obtained from three biological replicates (n = 3). Different letters within each column indicate significant differences, as determined by Fisher's LSD test (p < 0.05).
(DOCX)

**S10 Table. Effect of single, dual, and triple combinations of *Trichoderma* isolates on Southern blight disease in tomato plants caused by *Sclerotium rolfsii* in a field experiment at different weeks after transplanting (WAT).** In the treatments, SR denotes inoculation with the Southern blight pathogen *Sclerotium* rolfsii, while Tri2, Tri3, and Tri6 represent treatments with *Trichoderma* isolates Tri2, Tri3, and Tri6, respectively. In T9, treatment with the fungicide Provax-200 was included. Values (mean ± SE) for each treatment were obtained from three biological replicates (n = 3). Different letters within each column indicate significant differences, as determined by Fisher's LSD test (p < 0.05).
(DOCX)

## Acknowledgments

We sincerely wish to thank students and staff members in the Department of Plant Pathology, BSMRAU, for their support and assistance in the research work.

## Author contributions

**Conceptualization:** M. Motaher Hossain.

**Data curation:** Md. Robiul Hasan, Jannatun Nayeema.

**Investigation:** Nusrat Jahan Mishu, Md. Robiul Hasan.

**Supervision:** Shah Mohammad Naimul Islam, M. Motaher Hossain.

**Writing – original draft:** Nusrat Jahan Mishu, Md. Robiul Hasan, Jannatun Nayeema.

**Writing – review & editing:** Shah Mohammad Naimul Islam, M. Motaher Hossain.

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
