## [Decision Letter · Decision Letter 0]

PONE-D-25-26745Synergistic effects of Trichoderma isolates for enhancing growth, suppressing southern blight and modulating plant defense enzymes in tomatoPLOS ONE

Dear Dr. Hossain,

Thank you for submitting your manuscript to PLOS ONE. After careful consideration, we feel that it has merit but does not fully meet PLOS ONE’s publication criteria as it currently stands. Therefore, we invite you to submit a revised version of the manuscript that addresses the points raised during the review process.

We look forward to receiving your revised manuscript.

Kind regards,

Debasis Mitra

Academic Editor

PLOS ONE

Journal Requirements:

Reviewers' comments:

Reviewer's Responses to Questions

**Comments to the Author**

1. Is the manuscript technically sound, and do the data support the conclusions?

Reviewer #1: Yes

Reviewer #2: Yes

Reviewer #3: Yes

2. Has the statistical analysis been performed appropriately and rigorously? 

Reviewer #1: Yes

Reviewer #2: Yes

Reviewer #3: Yes

3. Have the authors made all data underlying the findings in their manuscript fully available?

Reviewer #1: Yes

Reviewer #2: Yes

Reviewer #3: Yes

4. Is the manuscript presented in an intelligible fashion and written in standard English?

Reviewer #1: Yes

Reviewer #2: Yes

Reviewer #3: Yes

5. Review Comments to the Author

Reviewer #1: 1. Figure resolution is very low, even though I downloaded the original figures, the text and many elements are blurry.

2. The Introduction is generally clear, but the study’s novelty and global significance are understated. After noting that “several studies” have been done in Bangladesh, please indicate whether comparable work has been reported elsewhere and explain how this manuscript advances the field.

3. In the Methods section, clarify the rationale for each experimental selection (isolate choice, inoculum rate, assay timing, etc.). Specify whether these choices follow prior literature or arise from preliminary experiments.

4. Authors discussed the mechanism could be “extracellular enzyme production, inhibition of pathogenicity factors, and enhanced antioxidative responses”. But how Tris caused these are still unclear (the fundamental mechanism).

5. “Synergistic” appears in the title and throughout the text, but the manuscript does not clearly demonstrate synergy. Provide quantitative evidence (e.g., interaction contrasts or synergy indices) or revise the terminology to reflect additive effects instead of synergy.

Reviewer #2: The manuscript is very relevant in the aspect of sustainable agriculture. The study shows the efficacy of various Trichoderma isolates for their potential to promote plant growth and to control the southern blight in tomato crops which is caused by Sclerotium rolfsii. The research is very comprehensive, combining in vitro, greenhouse and field trails and includes biochemical and enzymatic assays to elucidate underlying plant defense mechanisms. The molecular characterization of isolates along with the physiological and biochemical profiling is integrated appropriately in the study. Additionally, the correlation between enhanced growth and reduced oxidative damage under pathogen stress is also well supported by the data on H2O2, MDA and enzymatic activity level.

Following sentence in the abstract: “Trichoderma-treated plants challenged with S. rolfsii exhibited significantly reduced oxidative stress, as indicated by lower hydrogen peroxide (H₂O₂) and malondialdehyde (MDA) levels” could be more concise by rewriting it as “Trichoderma-treated plants challenged with S. rolfsii showed reduced oxidative stress, evidenced by lower H₂O₂ and MDA levels.”

Reviewer #3: This manuscript presents a comprehensive and methodically robust investigation into the synergistic potential of Trichoderma isolates in enhancing tomato growth and suppressing southern blight caused by Sclerotium rolfsii. The work includes both in vitro and in vivo experiments (seed tray, pot, and field), integrates biochemical and physiological assessments, and contributes meaningful insights into the mechanisms of Trichoderma-mediated biocontrol.

The findings are scientifically significant and align with growing interest in sustainable and eco-friendly plant disease management strategies. However, the manuscript would benefit from moderate revisions to improve readability, condense redundancies, and clarify some experimental details.

Abstract

The authors should provide 1-2 points of numerical data for key outcomes (e.g. significantly greater height, chloropyll content etc). When authors state that a result was significant, do they mean statistically significant? If so, insert a P value.

Introduction

Line 46 – “Tomato is a popular vegetable worldwide consumed”…please fix grammar here.

Line 47 – italicise Solanaceae

Line 89 – was there a third reference that was mistakenly deleted here. If so, please revise.

Materials & Methods

Line 124 -how many soil samples were collected in total from each plant?

Line 136 – why was the soil autoclaved twice? Would this affect the nutrient content? How do you know that spore-forming bacteria didn’t survive the process?

Line 140 – add supplier and supplier code for streptomycin sulfate and any additional chemicals used in this study.

Results

The results section is very long. The authors could condense or summarise the overlapping findings more clearly. For example, some findings are repeated across multiple subsections (e.g. growth promotion metrics in trays, pots and fields are summarised individually without describing overlaps).

6. PLOS authors have the option to publish the peer review history of their article (what does this mean? ). If published, this will include your full peer review and any attached files.

**Do you want your identity to be public for this peer review?** For information about this choice, including consent withdrawal, please see our Privacy Policy .

Reviewer #1: **Yes: ** Mairui Zhang

Reviewer #2: No

Reviewer #3: **Yes: ** Leonard Koolman

---

## [Author Response · Author response to Decision Letter 1]

20 Jun 2025

Rebuttal letter

Response to Editor

Thank you for the opportunity to revise our manuscript. We appreciate the constructive feedback from all the reviewers. I am pleased to confirm that I have thoroughly revised the manuscript and addressed all the comments provided by the reviewers. Additionally, we have fulfilled the journal's additional requirements, which have been indicated in the cover letter. Below are our responses to the specific comments from the reviewers.

Response to Reviewers

Reviewer #1:

Thank you for your constructive and valuable comments. We have agreed and addressed all of your comments in the revised manuscript. Please see our point-to-point responses to the specific comment below:

1. Figure resolution is very low, even though I downloaded the original figures, the text and many elements are blurry.

Response: Thank you for the suggestion. We have improved the figure resolution. Additionally, figures have been checked by using the Preflight Analysis and Conversion Engine (PACE) digital diagnostic tool, https://pacev2.apexcovantage.com/ to ensure that figures meet PLOS requirements.

2. The Introduction is generally clear, but the study’s novelty and global significance are understated. After noting that “several studies” have been done in Bangladesh, please indicate whether comparable work has been reported elsewhere and explain how this manuscript advances the field.

Response: Thank you for your insightful comment. Following your advice, we have carefully revised the Introduction to better highlight the novelty and broader significance of our study (L131-138).

3. In the Methods section, clarify the rationale for each experimental selection (isolate choice, inoculum rate, assay timing, etc.). Specify whether these choices follow prior literature or arise from preliminary experiments.

Response: Thank you for the valuable suggestion. We have revised the Methods section to include justifications for each experiment.

4. Authors discussed the mechanism could be “extracellular enzyme production, inhibition of pathogenicity factors, and enhanced antioxidative responses”. But how Tris caused these are still unclear (the fundamental mechanism).

Response: Thank you for the thoughtful comment. We agree with your observation and thus, have revised the Discussion section to clarify the underlying mechanisms by which Trichoderma isolates trigger biocontrol effects (L1012-1022).

5. “Synergistic” appears in the title and throughout the text, but the manuscript does not clearly demonstrate synergy. Provide quantitative evidence (e.g., interaction contrasts or synergy indices) or revise the terminology to reflect additive effects instead of synergy.

Response: Thank you for pointing this out. We agree with you and have revised the terminology throughout the manuscript, including the title, to reflect “additive” effects rather than “synergistic.”

Reviewer #2: The manuscript is very relevant in the aspect of sustainable agriculture. The study shows the efficacy of various Trichoderma isolates for their potential to promote plant growth and to control the southern blight in tomato crops which is caused by Sclerotium rolfsii. The research is very comprehensive, combining in vitro, greenhouse and field trails and includes biochemical and enzymatic assays to elucidate underlying plant defense mechanisms. The molecular characterization of isolates along with the physiological and biochemical profiling is integrated appropriately in the study. Additionally, the correlation between enhanced growth and reduced oxidative damage under pathogen stress is also well supported by the data on H2O2, MDA and enzymatic activity level.

Response: We appreciate your constructive and valuable comments. We have addressed all your suggestions in the revised manuscript. Please see our detailed point-to-point responses to each specific comment below:

1. Following sentence in the abstract: “Trichoderma-treated plants challenged with S. rolfsii exhibited significantly reduced oxidative stress, as indicated by lower hydrogen peroxide (H₂O₂) and malondialdehyde (MDA) levels” could be more concise by rewriting it as “Trichoderma-treated plants challenged with S. rolfsii showed reduced oxidative stress, evidenced by lower H₂O₂ and MDA levels.”

Response: Thank you for the helpful suggestion. We have revised the sentence in the abstract as recommended to improve clarity and conciseness (L35-38).

Reviewer #3: This manuscript presents a comprehensive and methodically robust investigation into the synergistic potential of Trichoderma isolates in enhancing tomato growth and suppressing southern blight caused by Sclerotium rolfsii. The work includes both in vitro and in vivo experiments (seed tray, pot, and field), integrates biochemical and physiological assessments, and contributes meaningful insights into the mechanisms of Trichoderma-mediated biocontrol.

The findings are scientifically significant and align with growing interest in sustainable and eco-friendly plant disease management strategies. However, the manuscript would benefit from moderate revisions to improve readability, condense redundancies, and clarify some experimental details.

Response: We sincerely thank the reviewer for the encouraging and constructive feedback. We appreciate the recognition of our work’s scientific significance and methodological rigor. In response to your suggestions, we have revised the manuscript to improve readability, removed redundancies, and clarified experimental details as recommended.

1. Abstract

The authors should provide 1-2 points of numerical data for key outcomes (e.g. significantly greater height, chloropyll content etc). When authors state that a result was significant, do they mean statistically significant? If so, insert a P value.

Response: Thank you for the helpful suggestion. We have revised the abstract to include key numerical data for various parameters and representative P values.

2. Introduction

Line 46 – “Tomato is a popular vegetable worldwide consumed”…please fix grammar here.

Response: The sentence has been revised for clarity and correctness (L54-55).

3. Materials & Methods

Line 124 -how many soil samples were collected in total from each plant?

Response: Thank you for the query. A total of three rhizospheric soil samples were collected from each plant in each location. This clarification has been added to the revised manuscript (147-149).

4. Line 136 – Why was the soil autoclaved twice? Would this affect the nutrient content? How do you know that spore-forming bacteria didn’t survive the process?

Response: Thank you for your insightful questions. The soil was autoclaved twice on consecutive days to effectively eliminate persistent microorganisms, including spore-forming bacteria. This is a commonly used method to minimize microbial interference in controlled experiments. To our observation, autoclaving does not significantly affect the levels of nitrogen (N), phosphorus (P), potassium (K), or organic matter in our soil.

5. Line 140 – add supplier and supplier code for streptomycin sulfate and any additional chemicals used in this study.

Response: We have added the information (L165).

6. Results

The results section is very long. The authors could condense or summarise the overlapping findings more clearly. For example, some findings are repeated across multiple subsections (e.g. growth promotion metrics in trays, pots and fields are summarised individually without describing overlaps).

Response: Thank you for the helpful suggestion. We have extensively revised the Results section to condense overlapping content and summarize key findings more clearly (L562-666).

---

## [Decision Letter · Decision Letter 1]

Additive effects of Trichoderma isolates for enhancing growth, suppressing southern blight and modulating plant defense enzymes in tomato

PONE-D-25-26745R1

Dear Dr. Hossain,

We’re pleased to inform you that your manuscript has been judged scientifically suitable for publication and will be formally accepted for publication once it meets all outstanding technical requirements.

Kind regards,

Dr. Debasis Mitra

Academic Editor

PLOS ONE

Additional Editor Comments (optional):

Reviewers' comments:

Reviewer #3: Thank you for revising the manuscript and addressing all my comments as well as the additional reviewer comments.

---

## [Editor Report · Acceptance letter]

PONE-D-25-26745R1

PLOS ONE

Dear Dr. Hossain,

I'm pleased to inform you that your manuscript has been deemed suitable for publication in PLOS ONE. Congratulations! Your manuscript is now being handed over to our production team.

Kind regards,

on behalf of

Dr. Debasis Mitra

Academic Editor

PLOS ONE